# SCALABLE KERNELS FOR GRAPHS WITH CONTINUOUS NODE ATTRIBUTES

## ABSTRACT

We introduce the Neighborhood Subgraph Pairwise Path Kernel (NSPPK), a scalable and interpretable graph kernel for node-attributed graphs. NSPPK compares neighborhoods connected through unions of shortest paths and directly integrates continuous node features without discretization. This yields explicit, sparse embeddings where graph similarities reduce to a single dot product. Feature extraction scales near-linearly in $|V|$, parallelizes efficiently, and is fully deterministic. Across six benchmarks with continuous attributes, NSPPK achieves the best average rank among graph kernels and frequently matches or outperforms modern GNNs—without any training or hyperparameter tuning. By combining scalability, interpretability, and expressive power, NSPPK offers a practical alternative for graph learning in low-data or reproducibility-critical settings. Its advantage lies in working robustly when data is scarce, yet scaling efficiently to hundreds of thousands of graphs when data is abundant.

## 1 INTRODUCTION

Graphs are a fundamental data structure for modeling relationships among entities, with applications in social networks (Newman, 2003), bioinformatics (Borgwardt et al., 2005), cheminformatics (Dobson & Doig, 2003), recommender systems (Ying et al., 2018), and cybersecurity (Huang et al., 2022). Unlike images or sequences embedded in regular grids, graphs capture irregular, non-Euclidean structures with variable neighborhoods and complex topologies (Bronstein et al., 2017). In many domains, nodes and edges carry attributes—categorical (e.g., atom types) or continuous (e.g., charges, coordinates, behavioral metrics).A central challenge in graph learning is how to compare such rich structures both effectively and efficiently. Two main families of methods have emerged. *Graph kernels* provide a classical and well-founded approach: they decompose graphs into substructures and measure similarity through carefully designed comparisons. Kernels are deterministic, interpretable, and often perform well in low-data settings. However, **most classical kernels assume discrete labels**, relying on exact matches. Applied to continuous data, they typically require discretization (Neumann et al., 2016b), which discards fine-grained information and may distort similarity. Empirically, kernels that integrate continuous features directly (Feragen et al., 2013b) outperform those based on discretization, but many variants still struggle with scalability, especially on larger graphs.In contrast, *Graph Neural Networks (GNNs)* (Kipf & Welling, 2017b; Xu et al., 2019b) naturally process continuous attributes and have achieved strong benchmark performance. Yet they usually demand large labeled datasets, intensive training, and extensive hyperparameter tuning, while their internal representations remain difficult to interpret (Errica et al., 2020; Hu et al., 2020b). These drawbacks limit their applicability in low-data regimes or in settings where reproducibility and transparency are critical.This trade-off motivates the search for approaches that combine the sample-efficiency and interpretability of kernels with the expressive power and flexibility of neural methods. Several recent kernels have moved in this direction by incorporating continuous features through embeddings (Feragen et al., 2013b), propagation (Neumann et al., 2016a), or WL-style extensions (Shervashidze et al., 2009a; Rieck et al., 2019). While more expressive, these methods often face scalability challenges, leaving room for further improvement.

**Our Contribution** We introduce the *Neighborhood Subgraph Pairwise Path Kernel* (NSPPK), a new graph kernel designed to combine scalability, interpretability, and support for continuous node attributes. Our method builds on the *Neighborhood Subgraph Pairwise Distance Kernel*

(NSPDK) (Costa & De Grave, 2010), a well-known kernel that compares fixed-radius neighborhoods around pairs of nodes. While NSPDK has proven effective in capturing structural information, it is limited to discrete labels and cannot directly exploit real-valued node features.

NSPPK extends NSPDK in three key ways: **Kernel design.** We replace fixed-radius neighborhoods alone with unions of shortest-path neighborhoods between node pairs, capturing dependencies that go beyond the reach of classical NSPDK features. **Continuous attributes.** Real-valued node (and edge) features are integrated directly into the kernel without discretization, preserving fine-grained information that would otherwise be lost. **Efficiency.** NSPPK yields explicit, sparse graph-level embeddings. Kernel evaluation reduces to a single dot product in $O(|E|)$ time, and feature extraction scales near-linearly in $|V|$, is trivially parallelizable, and requires only a few integer hyperparameters.

**Empirical results.** Across six benchmarks with continuous attributes, NSPPK attains the best average rank among graph kernels and often matches or outperforms GNN baselines, all without any training, hyperparameter tuning, or randomness.

## 2 RELATED WORK

Most graph kernels follow the *R-convolution* framework (Haussler, 1999), which decomposes structured objects into substructures and sums kernel evaluations. Examples include the graphlet kernel (Shervashidze et al., 2009b), Weisfeiler–Lehman (WL) subtree kernel (Shervashidze et al., 2011a), and NSPDK (Costa & De Grave, 2010). WL kernels are powerful but limited by the 1-WL test, while NSPDK counts fixed-radius neighborhoods around node pairs. To handle continuous attributes, early kernels such as marginalized random walk (Kashima et al., 2003; Gärtner et al., 2003; Vishwanathan et al., 2010) and subgraph-matching (Kriege & Mutzel, 2012) are expressive but computationally heavy. Propagation kernels (Neumann et al., 2016a) scale efficiently but rely on discretization. Shortest-path-based kernels (Borgwardt & Kriegel, 2005; Feragen et al., 2013b) capture long-range structure but suffer from high complexity. Recent work relaxes exact label matches via optimal transport, e.g., Wasserstein WL (Togninalli et al., 2019) and fused Gromov–Wasserstein (Vayer et al., 2019), though at high cost. Hybrid approaches integrate kernels with neural models, such as Deep Graph Kernels (Yanardag & Vishwanathan, 2015) and Graph Neural Tangent Kernels (Du et al., 2019). NSPPK builds on NSPDK but introduces two key innovations: (i) unions of shortest-path neighborhoods capture richer multi-scale dependencies, and (ii) continuous attributes are integrated directly without discretization. Unlike graph invariant kernels (Orsini et al., 2015), NSPPK avoids explicit subgraph matching, and its explicit embeddings allow $O(|E|)$ similarity computation while retaining interpretability.Recent graph kernels have sought to handle continuous node attributes by comparing *distributions* of node- or substructure-level representations,rather than relying on discrete label matching. MMD-GK (Sun & Fan, 2024) represents each graph as a distribution over node embeddings obtained via Laplacian smoothing of node features and measures similarity via maximum mean discrepancy (MMD) between these distributions. As this comparison is performed implicitly through pairwise kernel evaluations over nodes, MMD-GK does not produce an explicit, sparse graph-level feature map as in substructure based graph kernels.The Sliced Wasserstein Weisfeiler–Lehman (SWWL) kernel (Carpintero Perez et al., 2024) extends WL-style aggregation to continuous attributes by comparing distributions of node embeddings using sliced Wasserstein distances. SWWL achieves favorable scalability by replacing full optimal transport with random projections and quantile embeddings, but still represents each graph as a global distribution of node features, thereby discarding explicit structural subgraph correspondence and yielding dense graph-level embeddings. Most closely related to path-based methods, the Distributional Shortest-Path (DSP) graph kernel (Ye et al., 2025) augments classical shortest-path kernels by learning embeddings of shortest paths via neural language models. DSP captures both within-graph and dataset-wide distributional information by combining Transformer-based path embeddings, a partition kernel, and kernel mean embeddings. While highly expressive, DSP relies on learned substructure embeddings, repeated partitioning or clustering of node representations, and dense RKHS feature maps, resulting in substantial computational and memory overhead.In contrast to these distributional kernels, NSPPK constructs an *explicit* and sparse feature map based on combinatorial subgraphs defined by pairs of local neighborhoods connected through unions of shortest paths. Continuous node attributes are integrated deterministically by aggregating node attribute values within each hashed structural bucket, and graph similarity reduce to a single sparse dot product. This avoids neural representation learning, optimal transport, MMD, and

dataset-wide partitioning, while retaining strong discriminative power and near-linear complexity in the number of edges under small radii and degree cutoffs.

## 3 DEFINITIONS

A *graph* is a pair $G = (V, E)$, where $V$ is a finite set of vertices (or nodes) and $E \subseteq V \times V$ is a set of edges connecting pairs of vertices. A *labeled graph* is a graph $G = (V, E)$ equipped with a labeling function $\ell : V \cup E \to \Sigma$ that assigns each vertex and edge a label from a discrete alphabet $\Sigma$. An *attributed graph* is a graph $G = (V, E)$ endowed with an attribute function $f : V \cup E \to \mathbb{R}^d$ that assigns each vertex and edge a $d$-dimensional real-valued feature vector.

For a vertex $v \in V$, the *degree* of $v$ is the number of edges incident to it, $\deg(v) = |\{u \in V \mid (v, u) \in E\}|$, and its *(immediate) neighborhood* is $N(v) = \{u \in V \mid (v, u) \in E\}$. Optionally, a degree cutoff parameter $\tau$ can be introduced, restricting neighborhood expansions to $\min(\deg(v), \tau)$.

A *path* in $G$ is a sequence of vertices $(v_1, v_2, \ldots, v_k)$ such that $(v_i, v_{i+1}) \in E$ for all $1 \leq i < k$. The *length* of the path is the number of edges in the sequence, i.e., $k - 1$.

A *shortest path* from $v$ to $u$ is a path with the smallest possible length among all paths connecting $v$ and $u$. The *distance* between $v$ and $u$, denoted $d(v, u)$, is the length of a shortest path between them; if no path exists, $d(v, u)$ is defined to be infinite.

The *union of shortest paths* between vertices $v$ and $u$, denoted $U(v, u)$, is the subgraph consisting of all vertices and edges that belong to at least one shortest path from $v$ to $u$ (i.e., the union over all equally-short paths).

The *r-hop neighborhood* of a vertex $v$, denoted $N_r(v)$, is the set of vertices whose distance from $v$ is at most $r$, namely $N_r(v) = \{u \in V \mid d(v, u) \leq r\}$. Similarly, the $r$-hop neighborhood of a subgraph $S \subseteq G$ is the subgraph induced by all vertices $u \in V$ such that $\exists w \in S$ with $d(w, u) \leq r$.

*Anchors and connector path.* Given an (unordered) anchor pair $\{u, v\} \subseteq V$ with $u \neq v$ and distance $d(u, v)$, define the *connector path* of radius $r' \geq 0$ by $C_{r'}(u, v) := N_{r'}\big(U(u, v)\big)$, where $U(u, v)$ is the union of all shortest $u \leftrightarrow v$ paths (as defined above). Thus $C_0(u, v) = U(u, v)$ (only path nodes/edges), while $r' > 0$ "thickens" the connector by including all vertices within $r'$ hops of $U(u, v)$ (induced subgraph). If $u$ and $v$ are disconnected, set $C_{r'}(u, v) = \varnothing$ and ignore the pair. Unless stated otherwise, anchor pairs are *unordered* to avoid double counting.

**Notation summary.** Unless otherwise specified, we denote by $|V|$ and $|E|$ the numbers of vertices and edges, respectively; $K = \max_{v \in V} \deg(v)$ is the maximum degree and $\tau$ an optional degree cutoff. Distances are $d(u, v)$, $N_r(v)$ is the $r$-hop neighborhood of $v$, $U(u, v)$ the union of all shortest $u \leftrightarrow v$ paths, and $C_{r'}(u, v) = N_{r'}(U(u, v))$ the connector of radius $r'$. Lowercase $r, d, r'$ denote per-feature radii and distances, while uppercase $R, D, r'$ are their maximal values. The resulting feature vector for a graph $G$ under parameters $\theta = (R, D, r')$ is $f_G^\theta$.

## 4 METHOD

A widely used strategy for defining kernels between structured objects is to decompose them into constituent *substructures* and compare all possible substructure pairs using a base kernel. Kernels designed this way fall under the *R-convolution* framework (Haussler, 1999), which includes most classical graph kernels.

The Neighborhood Subgraph Pairwise Distance Kernel (NSPDK) (Costa & De Grave, 2010) instantiates this framework by counting pairs of fixed-radius neighborhoods at a given distance. However, NSPDK has two main limitations: (i) it only supports discrete node labels, and (ii) it uses only fixed-radius neighborhoods, missing richer structural patterns.

We propose the *Neighborhood Subgraph Pairwise Path Kernel* (NSPPK), which extends NSPDK in three ways:

1. A scalable, parallel graph kernel whose feature extraction runs in *near-linear* time in $|V|$ for fixed $(R, D)$ (and optional degree cap $\tau$), yielding explicit sparse embeddings so similarities reduce to a single $O(|E|)$ dot product.

2. A new feature family that pairs symmetric $r$-hop anchor neighborhoods $N_r(u), N_r(v)$ with a union-of-shortest-path connector $C_{r'}(u, v) = N_{r'}(U(u, v))$, capturing long-range topological interactions (with $r' = 0$ recovering the bare shortest-path union).

3. A principled integration of continuous node (and optionally edge) attributes directly into the hashing/aggregation pipeline—no discretization—preserving fine-grained information in deterministic, interpretable features.

The complete NSPPK feature set is obtained by enumerating all parameter configurations:

$$r_u, r_v \in \{0, \ldots, R\}, \quad d \in \{0, \ldots, D\}, \quad r' \in \{0, \ldots, R'\} \cup \{\varnothing\},$$

where $R, D, R'$ are small positive integers chosen for tractability. We denote by $r'$ the connector radius for any given feature and by $R'$ the maximal connector radius considered during extraction, so $r' \in \{0, \ldots, R'\}$.

## 4.1 NSPPK DEFINITION

Let $\theta = (R, D, R')$ denote the maximal radii and distances for feature extraction. For a graph $G$, let $f_G^\theta$ be the vector counting occurrences of each subgraph pattern in the NSPPK family. The kernel between $G$ and $G'$ is $k_\theta(G, G') = {f_G^\theta}^\top f_{G'}^\theta$. Because NSPPK features are defined per node, this can be written as $f_G^\theta = \sum_{v \in V} f_{G,v}^\theta$, where $f_{G,v}^\theta$ counts only features in which $v$ is one of the neighborhood centers or path endpoints.

## 4.2 FEATURE HASHING PIPELINE

We represent each subgraph pattern by a unique integer in $\{0, \ldots, 2^n - 1\}$ using a hierarchy of hash functions. This provides constant-time indexing into the feature vector and avoids explicit subgraph isomorphism checks.

**Base hash functions.** For any element $x$, $H_n(x) = \mathrm{sha256}(x) \bmod 2^n$ is the $n$-bit base hash. From $H_n$ we define: - $H^q(I)$: *sequence hash* of an ordered tuple $I = (x_1, \ldots, x_k)$, - $H^t(S)$: *multiset hash* of $S$, computed after lexicographic sorting to ensure order invariance.

**Node hash.** For each node $v$ (labels and neighborhoods as in Section 3) we set $N_h(v) = H^t\big(\{\, H^q([H_n(\ell(u)), H_n(\ell(e_{v,u}))]) : u \in N(v) \,\}\big)$ and $N_H(v) = H^q([H_n(\ell(v)), N_h(v)])$.

**Rooted graph hash.** For radius $r$, set $C_j^v = H^t(\{\, N_H(u) : u \in D_j^v \,\})$ with $D_j^v = \{\, u \mid d(v, u) = j \,\}$ and $G_H^r(v) = H^q([C_0^v, C_1^v, \ldots, C_r^v])$.

**Neighborhood pair hash.** For nodes $u, v$ at distance $d$ we compute $P_H^{r_u, r_v, d}(u, v) = H^q([H_n(d), H^t(\{G_H^{r_u}(u), G_H^{r_v}(v)\})])$.

**Union-of-shortest-paths hash.** Let $U(u, v)$ be the union of all shortest paths between $u$ and $v$. For each $j \in \{0, \ldots, d\}$ we set $\overline{C_{j,r}^v} = H^t(\{\, G_H^r(w) : w \in D_j^v \,\})$ and $U_H^{r,d}(u, v) = H^q([\overline{C_{0,r}^v}, \ldots, \overline{C_{d,r}^v}])$.

**Final feature vector.** The NSPPK vector $f_G^\theta$ is the histogram of all $P_H$ and $U_H$ hash values from $G$.

## 4.3 NODE ATTRIBUTE INTEGRATION

For graphs with continuous node attributes $A \in \mathbb{R}^{n \times p}$, let $F \in \mathbb{R}^{n \times f}$ be the binary node–feature incidence matrix, where $f = 2^n$ is the number of hash buckets. We compute $x = \mathrm{vec}(A^\top F) \in \mathbb{R}^{p \cdot f}$ so that each feature index stores the sum of attributes of all nodes in subgraphs contributing to that feature. Node weights can be incorporated by replacing $A$ with $\mathrm{diag}(w)A$, and the same approach extends to edge attributes.

## 4.4 WHY THE CONNECTOR PATH DISAMBIGUATES: AN ILLUSTRATIVE EXAMPLE

Figure 1 contrasts NSPDK and NSPPK features for two graphs that share the same $r=1$ anchor neighborhoods around $u$ and $v$ and the same distance $d(u,v)=5$, but differ in how $u$ and $v$ are connected. In the top row there is a *unique* shortest $u \leftrightarrow v$ path; in the bottom row there are *two* distinct shortest paths of equal length (their union forms a "ladder").

**NSPDK collapses the two cases.** NSPDK features only depend on $(N_r(u), d(u,v), N_r(v))$. Since $N_1(u)$, $N_1(v)$, and $d(u,v)$ are identical in both graphs, the NSPDK hash coincides: $\phi_{u,v}^{(r=1,d=5)} = \text{hash}(N_1(u), d(u,v), N_1(v))$, so NSPDK *cannot* distinguish them.

**NSPPK separates them.** NSPPK augments the pair of neighborhoods with the *connector* $C_{r'}(u,v) = N_{r'}(U(u,v))$, where $U(u,v)$ is the union of all shortest $u \leftrightarrow v$ paths. The structural feature becomes $\psi_{u,v}^{(r=1,d=5,r'=1)} = \text{hash}(N_1(u), C_1(u,v), N_1(v))$. In the top graph, $U(u,v)$ is a simple path; in the bottom graph, $U(u,v)$ contains two parallel shortest paths. Consequently $C_{r'}(u,v)$ differs (already for $r'=0$; $r'=1$ merely "thickens" the union), and the NSPPK hashes are distinct. This is precisely the extra resolution provided by the connector.

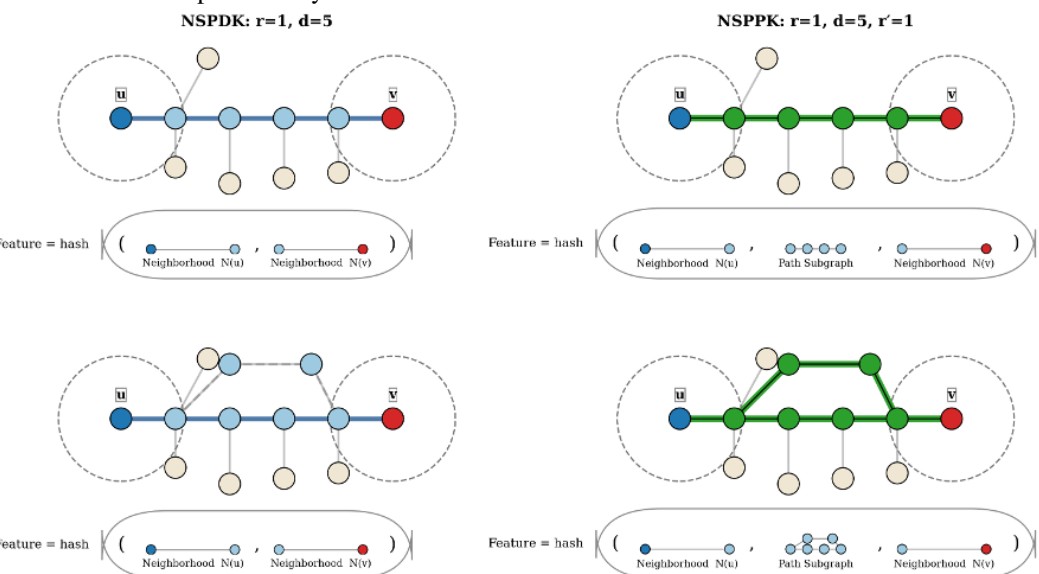

Figure 1: **NSPDK vs. NSPPK on anchors** $u, v$ **with** $r=1$, $d=5$. *Left:* NSPDK features for the case with a single shortest path (top) and two equal-length shortest paths (bottom). Because NSPDK uses only $(N_r(u), d(u,v), N_r(v))$, both cases produce the *same* feature. *Right:* NSPPK includes the connector $C_{r'}(u,v) = N_{r'}(U(u,v))$ (shown in green). The connector is a simple path in the top graph but a two-path union in the bottom graph, so NSPPK assigns *different* features (already for $r'=0$; here $r'=1$ is shown).

## 4.5 COMPLEXITY ANALYSIS

The main cost in NSPPK is extracting subgraphs via breadth-first search (BFS) up to depth $B = \max(R, D)$. A single BFS explores at most $O(K^B)$ vertices in the worst case (with $K = \max_{v \in V} \deg(v)$), and repeating this over all $|V|$ centers gives $O(|V|K^B)$. With a degree cutoff $\tau$, the branching factor becomes $K_{\text{eff}} = \min(K, \tau)$, yielding $O(|V|K_{\text{eff}}^B)$. Incorporating $d$-dimensional attributes adds only a multiplicative factor of $d$. Once features are extracted, kernel computation reduces to a sparse dot product $k(G, G') = f_G^\top f_{G'}$ with cost $O(\text{nnz}(f_G) + \text{nnz}(f_{G'}))$, scaling near-linearly with the number of edges $|E|$. In summary, under realistic settings where $K$ and $B$ are small (often $\lesssim 6$), NSPPK achieves near-linear scaling in $|V|$ (and thus $|E|$), with attribute integration adding only a linear factor in $d$.

## 4.6 GRAPH ISOMORPHISM

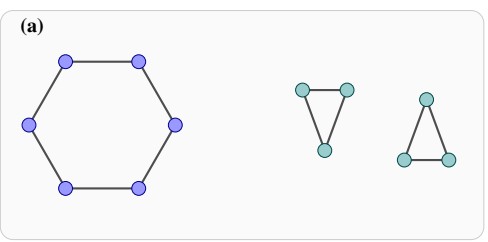 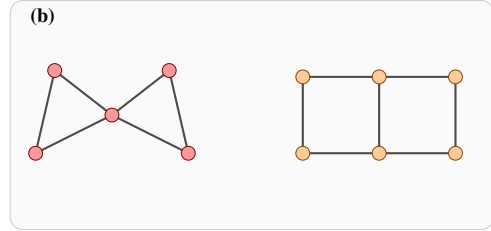

Figure 2: Two 1–WL/MP–GNN indistinguishable pairs that NSPPK separates. **(a)** $C_6$ (6 nodes-cycle graph) vs. $K_3 \cup K_3$ (two disconnected triangles). **(b)** Bow-tie vs. $2 \times 3$ grid ("ladder").

Distinguishing non-isomorphic graphs is essential for expressive graph kernels. Two graphs $G = (V_G, E_G)$ and $H = (V_H, E_H)$ are *isomorphic*, denoted $G \cong H$, if a bijection $\phi : V_G \to V_H$ preserves adjacency. While Graph Isomorphism (GI) resides in a nuanced complexity class (recently shown to admit a quasi-polynomial-time algorithm (Babai, 2016)), valid graph kernels must at least satisfy *isomorphism-invariance*. More valuable, though, is *isomorphism-discrimination*: ensuring $k(G, H) < k(G, G)$ for $G \ncong H$. The expressive power of many classical kernels and message-passing GNNs is limited by the *1-dimensional Weisfeiler–Lehman (1-WL)* test, which iteratively aggregates hashed neighborhood label multisets (Shervashidze et al., 2011a; Morris et al., 2019). If two graphs cannot be distinguished by 1-WL, neither can any derived WL kernel. Likewise, standard MP-GNNs align with 1-WL in distinguishing power (Xu et al., 2019b). Higher-order WL variants (like k-WL) and their GNN instantiations do surpass 1-WL but often suffer steep computational costs (Morris et al., 2019). Figure 2 illustrates graph pairs that are non-isomorphic yet indistinguishable by 1-WL or MP-GNNs due to identical local neighborhoods and degree sequences. NSPPK incorporates *shortest-path connectivity between rooted neighborhoods* by constructing features of the form $\psi(N_r(v), C_{r'}(v, u), N_r(u))$ for all anchor pairs $(v, u)$ at distance $d(v, u)$, where $C_{r'}(v, u) = N_{r'}(U(v, u))$ encodes the union of all shortest $v \leftrightarrow u$ paths. This allows NSPPK to distinguish graphs that are indistinguishable for 1-WL kernels and message-passing GNNs but differ in global connector structure, such as the pairs shown in Figure 2. Consequently, NSPPK achieves strictly higher discriminative power than 1-WL-based kernels and MP-GNNs, while remaining computationally efficient.

**NSPPK vs. NSPDK.** As formalised in Proposition C in the Appendix, NSPPK is strictly more expressive than NSPDK under any fixed finite anchor-radius budget. Specifically, there exists an infinite family of non-isomorphic graph pairs $(G_n, H_n)$ and a constant $R_0 \geq 1$ such that NSPDK produces identical feature vectors for all parameter choices $(R, D)$ with $R \leq R_0$, whereas NSPPK with the same anchor radius $R \leq R_0$ and some connector radius $R' \geq 0$ assigns distinct feature vectors, assuming idealised, isomorphism-aware hashing. Figure 3 shows a concrete instance of such a pair. The two graphs are locally indistinguishable: all $r$-hop neighborhoods $N_r(v)$ match for every $r \leq R_0$, and all pairwise distances $d(u, v)$ are identical. Consequently, NSPDK—which only encodes pairs of local neighborhoods together with the distance between them—collapses the two graphs under any $(R, D)$ with $R \leq R_0$.

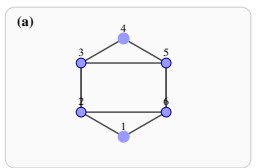 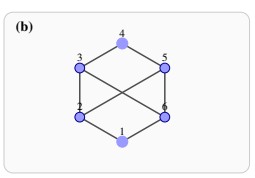

Figure 3: **NSPPK vs. NSPDK on a 6-cycle with distinct connectors.** Both graphs share the same outer 6-cycle and identical $r$-hop neighborhoods for all $r \leq R_0$ (here $R_0 = 1$), so NSPDK produces identical features for any $(R, D)$ with $R \leq R_0$. **(a)** Inner square on nodes $\{2, 3, 5, 6\}$. **(b)** Inner cross via diagonals $(2, 5)$ and $(3, 6)$. The union-of-shortest-paths connector between nodes 1 and 4 forms a cycle in (a) and two node-disjoint paths in (b). Under idealised isomorphism-aware hashing, NSPPK captures this connector structure and separates the graphs; NSPDK does not.

The difference lies solely in the connector region between opposite corners of the 6-cycle (nodes 1 and 4). In the *square* graph, the union of all shortest 1↔4 paths, $U(1, 4)$, induces a cycle; in the *cross* graph, $U(1, 4)$ consists of two internally node-disjoint diagonal paths forming an "X". Since NSPPK explicitly incorporates the connector $C_{r'}(1, 4) = N_{r'}(U(1, 4))$ into its feature representation, these two non-isomorphic connectors yield distinct feature hashes under idealised hashing.

### 4.7 BALANCING FEATURE SIZE AND HASH COLLISIONS

The number of hash bits directly determines the dimensionality of the feature space. Using fewer bits increases the probability of *hash collisions*—different substructures mapping to the same index—which introduces noise and can reduce predictive accuracy. Conversely, very large bit sizes (e.g., $n \geq 20$) yield millions of potential features, inflating dimensionality and memory usage. While sparse representations mitigate storage overhead, excessively large codomains may still become impractical for downstream learning models. To examine this trade-off, we trained a random forest classifier on 1,300 molecular graphs from PubChem AID 463230 (pPAFAH inhibition assay), excluding node attributes. As shown in Figure 4, predictive performance declines only gradually as the bit size decreases. In

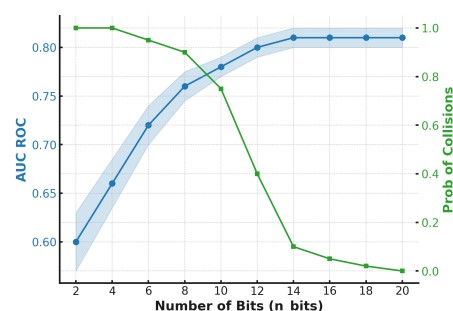

Figure 4: Predictive performance vs. number of hash bits.

particular, accuracy remains stable even at ~14 bits (about 16k features), despite collision rates exceeding 10%. This suggests that collisions involving infrequent features are largely tolerated by the model, enabling compact yet effective representations.

## 5 EXPERIMENTS

### 5.1 SMALL-SCALE DATASETS

We evaluate NSPPK against kernel and neural baselines on six node-attributed graph classification benchmarks (Nikolentzos et al., 2021; Errica et al., 2020): *ENZYMES* and *PROTEINS_full* (Borgwardt et al., 2005), *BZR* and *COX2* (molecular activity) (Sutherland et al., 2003), *Synthie* (Morris et al., 2016), and *SYNTHETICnew* (Feragen et al., 2013a). These cover biological, molecular, and synthetic graphs.

**Baselines.** Kernel methods: GraphHopper (GH) (Feragen et al., 2013b), Propagation Kernel (PK) (Neumann et al., 2016a), Subgraph Matching (SM) (Kriege & Mutzel, 2012), Multiscale Laplacian (ML) (Kondor & Pan, 2016), Shortest Path (SP) (Borgwardt & Kriegel, 2005), HSSPK-SP/WL (Morris et al., 2016), WWL (Togninalli et al., 2019), linearFGW (Nguyen & Tsuda, 2023), and NP (Fang et al., 2023), plus discretized NSPPK and WL. All kernels use SVM classifiers (LIBSVM (Chang & Lin, 2011)).

**Neural baselines**: DGCNN (Zhang et al., 2018), GraphSAGE (Hamilton et al., 2017), InfoGraph (Sun et al., 2019), GIN (Xu et al., 2019b), GraphCL (You et al., 2020), AttentiveFP (FNP) (Gasteiger et al., 2020), PNA (Corso et al., 2020), and PDF (Yang et al., 2023a).

**Protocol.** We follow the fair evaluation setup of (Errica et al., 2022): 10-fold cross-validation with 10% validation from training data. Kernels: SVM with $C$ tuned on validation; multiple hyperparameter configurations tested; kernel computation times reported for best models. GNNs: trained up to 1000 epochs with early stopping; tuned via validation. **Controls.** We also test (i) *attributes only*, where graphs are represented by summed node attributes, and (ii) *structure only*, where attributes are removed (Appendix J).

**NSPPK**: fixed configuration across all datasets ($R=1$, $D=4$, $R'=1$, 16-bit hashing); no per-dataset tuning. For comparison with kernels we use LIBSVM, and for GNNs we use NSPPK features with XGBoost.

**Runtime.** Kernel methods run on a single CPU core; GNNs run on CPUs with library-level multithreading. All experiments used a SLURM-managed cluster with NVIDIA A100 GPUs, Intel Xeon Gold 5317 CPUs (24 cores), and 64 GB RAM. Kernels exceeding 24h per fold are reported as timeouts.

**Results.** Against *kernels*, NSPPK achieves the best accuracy on three of six benchmarks (*SYNTHETICnew*, *BZR*, *COX2*) and the best overall average rank (2.25 vs. 3.50 for WWL). While it does not win every dataset, it consistently attains the strongest aggregate rank across kernel competitors, showing

broad reliability rather than isolated peaks. It scales reliably, unlike some kernels that time out on attribute-rich data. Against *neural networks*, NSPPK+XGBoost achieves the best accuracy on four datasets and the best overall rank (2.00), outperforming strong GNNs such as GIN, GraphCL, and PDF. Even where specific architectures edge out NSPPK on an individual dataset, it still delivers the top average rank across neural baselines, underscoring robustness across diverse tasks. Even in the structure-only setting, NSPPK remains competitive, indicating that it captures complementary structural and attribute information robustly across domains.

Table 1: Classification accuracy (%) **with** node attributes (+) with Avg Rank.

| Method | SYNTHETIC new | Synthie | BZR | COX2 | ENZYMES | PROTEINS | Avg Rank |
|---|---|---|---|---|---|---|---|
| SM | **TIMEOUT** | **TIMEOUT** | $83.96 \pm 3.85$ | $78.81 \pm 4.49$ | **TIMEOUT** | **TIMEOUT** | 8.75 |
| SP | **TIMEOUT** | **TIMEOUT** | **TIMEOUT** | **TIMEOUT** | **TIMEOUT** | **TIMEOUT** | – |
| ML | $50.33 \pm 9.00$ | $59.38 \pm 4.38$ | $82.96 \pm 5.24$ | $77.53 \pm 5.53$ | $36.33 \pm 4.33$ | $72.06 \pm 3.60$ | 11.08 |
| PK | $54.33 \pm 10.11$ | $71.75 \pm 6.43$ | $78.77 \pm 1.01$ | $78.16 \pm 8.07$ | $20.67 \pm 2.70$ | $59.70 \pm 0.16$ | 11.75 |
| HSPPK_WL | $60.33 \pm 6.74$ | $90.75 \pm 8.66$ | $85.67 \pm 3.46$ | $80.30 \pm 4.67$ | $60.17 \pm 6.26$ | $72.96 \pm 4.84$ | 5.08 |
| HSPPK_SP | $57.00 \pm 7.81$ | $91.25 \pm 3.01$ | $84.49 \pm 5.04$ | $80.31 \pm 5.70$ | $58.50 \pm 5.55$ | $68.55 \pm 5.24$ | 6.42 |
| GH | $77.33 \pm 7.42$ | $72.75 \pm 8.32$ | $85.94 \pm 5.17$ | $78.90 \pm 2.95$ | $66.50 \pm 6.17$ | $72.06 \pm 3.64$ | 5.42 |
| NP | **$99.00 \pm 2.13$** | $29.00 \pm 5.72$ | $86.18 \pm 5.53$ | $78.16 \pm 4.47$ | $43.00 \pm 5.76$ | $64.62 \pm 5.13$ | 7.83 |
| linearFGW-RAW | $62.00 \pm 8.46$ | $58.00 \pm 7.05$ | $76.34 \pm 5.68$ | $77.93 \pm 3.68$ | $56.67 \pm 7.07$ | $69.47 \pm 0.91$ | 11.58 |
| linearFGW-WL1 | $72.33 \pm 8.70$ | $73.00 \pm 5.34$ | $78.52 \pm 3.99$ | $72.13 \pm 7.40$ | $47.00 \pm 4.70$ | $59.66 \pm 0.38$ | 10.58 |
| linearFGW-WL2 | $71.33 \pm 7.48$ | $61.75 \pm 7.50$ | $78.51 \pm 2.98$ | $75.16 \pm 3.34$ | $41.00 \pm 7.57$ | $59.75 \pm 0.43$ | 11.50 |
| WL (disc.) | $88.33 \pm 5.63$ | $74.75 \pm 7.86$ | $83.71 \pm 4.83$ | $77.10 \pm 5.59$ | $50.67 \pm 7.03$ | $71.16 \pm 4.03$ | 7.58 |
| WLOA | $85.33 \pm 5.62$ | $75.50 \pm 10.50$ | $84.20 \pm 4.47$ | $74.54 \pm 5.40$ | $66.33 \pm 5.21$ | $71.16 \pm 1.97$ | 5.92 |
| WWL | $58.33 \pm 7.78$ | **$97.00 \pm 3.12$** | $86.45 \pm 4.46$ | $79.03 \pm 3.84$ | **$73.67 \pm 5.26$** | **$77.18 \pm 5.27$** | 3.50 |
| NSPDK (disc.) | $96.33 \pm 3.14$ | $83.75 \pm 4.64$ | $85.70 \pm 3.90$ | $80.30 \pm 4.15$ | $52.67 \pm 4.67$ | $72.87 \pm 1.51$ | 4.75 |
| **NSPPK (ours)** | **$99.00 \pm 1.52$** | $86.75 \pm 4.75$ | **$87.17 \pm 3.58$** | **$81.16 \pm 2.30$** | $60.50 \pm 5.38$ | $74.66 \pm 3.81$ | **2.25** |
| *Attributes only* | $54.33 \pm 9.55$ | $53.00 \pm 4.30$ | $78.77 \pm 1.01$ | $78.16 \pm 8.07$ | $55.67 \pm 5.38$ | $62.80 \pm 2.52$ | 11.25 |

*Note:* "Attributes only" participates in Avg Rank like any other method.

Table 2: Neural networks: classification accuracy (%) **with** node attributes (+) and Avg Rank.

| Method | SYNTHETIC new | Synthie | BZR | COX2 | ENZYMES | PROTEINS | Avg Rank |
|---|---|---|---|---|---|---|---|
| DGCNN | $46.67 \pm 5.63$ | $50.00 \pm 5.70$ | $79.40 \pm 3.32$ | $77.15 \pm 0.06$ | $33.33 \pm 9.37$ | $73.86 \pm 3.56$ | 9.00 |
| GraphSAGE | $76.67 \pm 7.70$ | $85.00 \pm 3.45$ | $83.70 \pm 4.44$ | $80.93 \pm 0.07$ | $65.00 \pm 5.73$ | $75.02 \pm 3.29$ | 4.33 |
| InfoGraph | $65.00 \pm 16.05$ | $85.75 \pm 8.50$ | $79.01 \pm 3.42$ | $77.77 \pm 14.20$ | $53.33 \pm 7.84$ | $66.37 \pm 6.25$ | 7.58 |
| GIN | $83.67 \pm 5.92$ | $97.50 \pm 2.50$ | $84.17 \pm 6.14$ | $81.80 \pm 6.14$ | $68.30 \pm 5.43$ | $62.10 \pm 5.26$ | 3.75 |
| GraphCL | $67.00 \pm 9.48$ | $78.75 \pm 7.18$ | $84.17 \pm 3.62$ | $80.34 \pm 6.95$ | $48.17 \pm 6.93$ | $75.82 \pm 2.73$ | 5.08 |
| GNN | $64.67 \pm 7.92$ | $85.00 \pm 5.92$ | $85.66 \pm 4.60$ | $79.92 \pm 7.08$ | $65.17 \pm 8.41$ | $66.57 \pm 6.10$ | 5.08 |
| FNP | $53.33 \pm 11.16$ | $36.00 \pm 8.60$ | $79.49 \pm 3.94$ | $78.22 \pm 7.01$ | $32.17 \pm 12.98$ | $70.17 \pm 3.15$ | 8.75 |
| PNA | $55.67 \pm 20.66$ | $92.50 \pm 3.71$ | $79.01 \pm 4.33$ | $78.22 \pm 7.02$ | $20.83 \pm 7.12$ | $75.11 \pm 3.60$ | 6.83 |
| PDF | $97.67 \pm 2.60$ | $64.25 \pm 7.50$ | $83.68 \pm 3.81$ | $82.23 \pm 7.00$ | $65.00 \pm 4.65$ | $72.14 \pm 4.48$ | 4.58 |
| NSPPK feat. (XGBoost) | **$98.67 \pm 2.21$** | $87.75 \pm 3.94$ | **$88.66 \pm 2.89$** | **$82.90 \pm 4.39$** | $60.17 \pm 6.30$ | **$77.37 \pm 4.80$** | **2.00** |
| *Attributes only* | $54.33 \pm 9.55$ | $53.00 \pm 4.30$ | $78.77 \pm 1.01$ | $78.16 \pm 8.07$ | $55.67 \pm 5.38$ | $62.80 \pm 2.52$ | 9.00 |

In the neural setting, the explicit NSPPK+XGBoost pipeline achieves an average runtime rank of $4.83$. It is clearly faster than heavyweight architectures such as GIN and DGCNN, while remaining competitive with mid-range models like FNP and PDF.

**Runtime analysis.** Detailed runtime tables are reported in Appendix I. Although it does not match the extreme speed of very lightweight self-supervised baselines (e.g., GraphCL, PNA), NSPPK+XGBoost simultaneously delivers the best accuracy overall, underlining its strong efficiency–accuracy trade-off. For comparability, kernel runtimes there are measured on a single CPU core, which also applies to NSPPK. Under this constraint, NSPPK is not the absolute fastest (average rank $6.33$), but it remains substantially more efficient than expressive kernels such as GH or HSPPK, while achieving higher accuracy. This positions NSPPK as a favorable compromise: slower than the simplest structure-only kernels, but far more accurate, and significantly faster than heavier graph kernels.

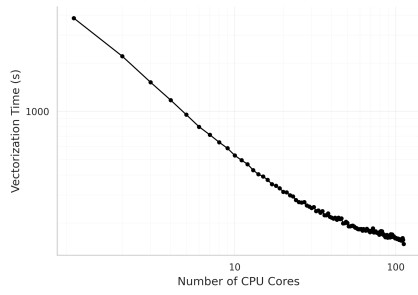

Figure 5: Vectorization time vs. CPU cores.

## 5.2 Ablation Study

Importantly, Figure 5 demonstrates that NSPPK is *parallelizable*: vectorization scales nearly linearly with the number of CPU cores. Thus, while our tables reflect conservative single-core timings for fairness, NSPPK can in practice achieve much faster wall-clock runtimes on multi-core systems.

NSPPK consistently achieves state-of-the-art accuracy without the need for dataset-specific training or hyperparameter tuning. Compared to existing kernels, it offers a substantially better balance between expressivity and efficiency, and when paired with XGBoost, it often surpasses neural baselines in predictive performance. While the runtime tables report conservative single-core measurements for fairness, Figure 5 shows that NSPPK scales nearly linearly with the number of CPU cores, enabling much faster wall-clock runtimes in practice.

We assessed the impact of three components of NSPPK: the distance feature, the union-of-shortest-paths connector, and the high-degree thresholding heuristic. All runs used fixed parameters ($R = 1$, $D = 4$, $r' = 1$) and a `RandomForestClassifier`.

Table 3: Ablation study: Accuracy (%) $\pm$ Std.

| Setting | ENZYMES | BZR | PROTEINS | SYNTHIE | SYNTHETICnew | COX2 |
|---|---|---|---|---|---|---|
| No Distance | $57.17 \pm 6.99$ | $85.69 \pm 5.52$ | $76.37 \pm 3.85$ | $69.00 \pm 6.24$ | $98.33 \pm 2.24$ | $81.16 \pm 2.45$ |
| No Path | $52.83 \pm 6.28$ | $86.98 \pm 3.72$ | $76.10 \pm 4.11$ | $79.81 \pm 1.53$ | $98.67 \pm 1.63$ | $80.73 \pm 2.30$ |
| No Threshold | $55.50 \pm 4.48$ | $87.42 \pm 2.25$ | $75.92 \pm 5.05$ | $79.00 \pm 5.15$ | $99.00 \pm 1.53$ | $81.81 \pm 3.78$ |
| **Full NSPPK** | $\mathbf{57.83 \pm 5.43}$ | $\mathbf{87.90 \pm 3.56}$ | $\mathbf{76.20 \pm 3.74}$ | $\mathbf{85.50 \pm 4.85}$ | $\mathbf{99.00 \pm 1.53}$ | $\mathbf{82.46 \pm 3.74}$ |

Results show that each component contributes depending on the dataset, with the full model consistently matching or exceeding the ablations. The degree-threshold heuristic, in particular, provides a robust improvement across datasets.

## 5.3 Sensitivity Analysis

We study how predictive performance varies with the anchor radius $R$, distance parameter $D$, and connector radius $r'$. On the COX2 dataset, we perform a stratified 10-fold cross-validation experiment, varying one parameter at a time while keeping the other two fixed at $(R, D, R') = (1, 4, 1)$. Table 4 reports the average accuracy and standard deviation. Across all three sweeps, accuracy remains very stable: performance fluctuates by only about 1–2 percentage points across the entire range. This matches intuition for small, sparse molecular graphs, where large radii tend to add redundant, quasi-global patterns rather than genuinely new substructures. In practice, NSPPK therefore does not require fine-grained tuning of $(R, D, R')$ on such datasets.

Table 4: Sensitivity of NSPPK accuracy to structural parameters on COX2 (stratified 10-fold cross-validation). Each column varies one hyperparameter while fixing the others at $(R, D, R') = (1, 4, 1)$.

| Value | Radius $R$ | Distance $D$ | Connector $r'$ |
|---|---|---|---|
| 0 | $0.81 \pm 0.05$ | $0.81 \pm 0.05$ | $0.81 \pm 0.05$ |
| 1 | $0.81 \pm 0.05$ | $0.81 \pm 0.05$ | $0.81 \pm 0.05$ |
| 2 | $0.80 \pm 0.03$ | $0.80 \pm 0.04$ | $0.79 \pm 0.03$ |
| 3 | $0.80 \pm 0.05$ | $0.80 \pm 0.05$ | $0.80 \pm 0.03$ |
| 4 | $0.80 \pm 0.03$ | $0.81 \pm 0.05$ | $0.80 \pm 0.04$ |
| 5 | $0.80 \pm 0.03$ | $0.80 \pm 0.04$ | $0.79 \pm 0.04$ |
| 6 | $0.79 \pm 0.03$ | $0.80 \pm 0.04$ | $0.80 \pm 0.04$ |

## 5.4 Larger-Scale Dataset Experiment

We further evaluated NSPPK on the large-scale `ogbg-molpcba` benchmark from the Open Graph Benchmark suite (Hu et al., 2020a), which contains 437,929 molecular graphs with node attributes and 128 binary classification tasks. Using a single fixed configuration ($R = 1$, $D = 4$, $R' = 1$, $n_{\text{bits}} = 16$), we computed explicit NSPPK features for the entire dataset in under one hour on CPU. Across all 128 tasks, NSPPK achieved an average validation AP of **0.2186** and an average test AP of **0.2079**, without any hyperparameter tuning or GPU acceleration. On the OGB leaderboard, state-of-the-art neural architectures such as Graphormer (Ying et al., 2021), PDF (Yang et al., 2023b), and HyperFusion (Zhang et al., 2024) achieve $\sim$0.30–0.32 test AP, typically relying on extensive pretraining, careful hyperparameter tuning, and GPU acceleration. Tuned mid-range models such as PNA (Corso et al., 2020), GIN (Xu et al., 2019a), and AttentiveFP (FNP) (Xiong et al., 2019) reach $\sim$0.25–0.30. By contrast, NSPPK attains 0.2079 test AP without any hyperparameter tuning, pretraining, or GPU usage, computing explicit features for all 438k graphs in under one hour

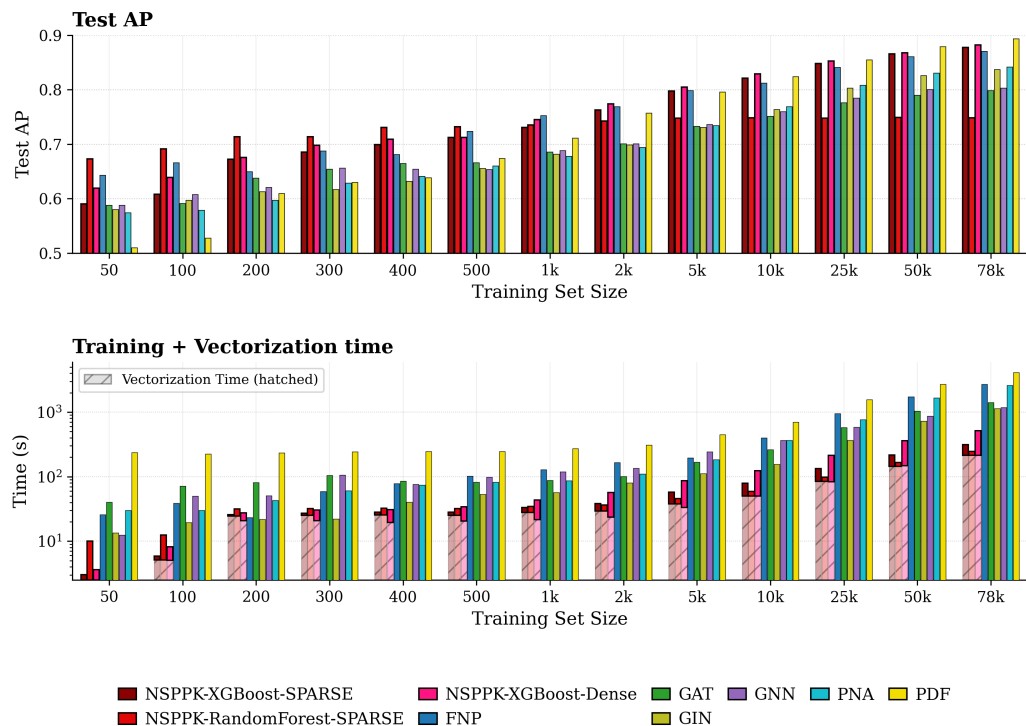

Figure 6: Learning curves on `ogbg-molpcba` (task 95). NSPPK (red) compared with neural baselines including GIN, GAT, PNA, PDF, and FNP.

on CPU.This positions NSPPK not as a replacement for the very best neural models, but as a complementary approach: it offers deterministic, training-free baselines that are highly competitive given their simplicity and efficiency. In practice, NSPPK fills a unique niche: when compute budgets are limited, when reproducibility is paramount, or when only small amounts of labeled data are available, it provides a strong, interpretable alternative that scales easily to hundreds of thousands of graphs.A case study on task 95 (Figure 6) shows that NSPPK exhibits strong sample efficiency: it outperforms neural baselines at small training sizes and remains substantially faster in the sparse variant. At scale, high-capacity models such as PDF eventually overtake NSPPK in absolute accuracy, but the gap remains modest given NSPPK's simplicity. Further implementation details and hyperparameter settings are provided in Appendix L.

## 6 CONCLUSION

We presented the Neighborhood Subgraph Pairwise Path Kernel (NSPPK), a scalable and interpretable extension of NSPDK that enriches neighborhood features with union-of-shortest-path connectors and integrates continuous node attributes without discretization. NSPPK produces explicit embeddings, enabling efficient, deterministic, and training-free similarity computation. Across six node-attributed benchmarks and a large-scale molecular dataset, NSPPK consistently outperforms classical kernels and often rivals or surpasses graph neural networks without training or hyperparameter tuning, providing strong baselines when compute, data, or reproducibility budgets are tight and complementing resource-intensive neural pipelines. While it is not the top performer on every dataset, it repeatedly secures the best overall ranks against both kernel and neural baselines, highlighting dependable, across-the-board strength. Its versatility spans low-data and large-scale regimes, maintaining predictable CPU-only runtimes and near-linear scalability in $|V|$ for transparent, easy-to-deploy solutions. In summary, NSPPK bridges classical kernel methods and neural approaches by favoring deterministic, efficient feature extraction over end-to-end training while retaining enough expressive power to stay competitive. Future work will explore hybrid kernel–neural models, automatic feature selection, and domain-specific adaptations.

ACKNOWLEDGEMENTS

For the purpose of open access, the authors have applied a Creative Commons Attribution (CC BY) licence to any Author Accepted Manuscript version arising from this submission.

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

APPENDIX

## A DATASETS

The graphs are undirected, with nodes labeled, attributed, or both. All datasets are publicly accessible Kersting et al. (2016); Morris et al. (2020) and have been widely used in comparative studies of graph kernels and GNNs Nikolentzos et al. (2021).

**BZR** consists of 405 chemical compounds represented as graphs, where nodes correspond to atoms and edges to chemical bonds. The task is to predict whether a compound acts as a ligand for the benzodiazepine receptor Dobson & Doig (2003).

**COX2** contains 467 molecules represented as graphs, labeled according to their activity against the cyclooxygenase-2 enzyme (COX-2 inhibitor classification) Dobson & Doig (2003).

**ENZYMES** consists of 600 protein tertiary structures from the BRENDA database. Each protein belongs to one of six top-level enzyme commission (EC) classes, and the task is to predict the correct class Borgwardt et al. (2005).

**PROTEINS** and **PROTEINS_full** represent proteins as graphs, where vertices correspond to secondary structure elements. Edges connect vertices that are adjacent in the amino acid sequence or in 3D space. The classification task is to distinguish enzymes from non-enzymes Borgwardt et al. (2005).

**SYNTHETICnew** contains 300 synthetic graphs evenly split into two classes. Each graph has 100 vertices and 196 edges with normally distributed node attributes. Class 1 graphs are generated by rewiring 5 edges and permuting 10 node attributes; Class 2 graphs by rewiring 10 edges and permuting 5 attributes. Gaussian noise is added to all attributes Shervashidze et al. (2011b).

**SYNTHIE** contains 400 synthetic graphs across four classes, each with 15 real-valued node attributes. Graphs are constructed from perturbed Erdős–Rényi base graphs and combined with two distinct attribute distributions Morris et al. (2016).

Table 5: Dataset statistics.

| Dataset | #Graphs | #Classes / Tasks | Avg. |V| | Avg. |E| | Avg. deg. | Max deg. | Attr. dim. |
|---|---|---|---|---|---|---|---|
| BZR | 405 | 2 | 35.75 | 38.36 | 2.15 | 4 | 3 |
| COX2 | 467 | 2 | 41.23 | 43.45 | 2.11 | 4 | 3 |
| ENZYMES | 600 | 6 | 32.63 | 62.14 | 3.86 | 9 | 18 |
| PROTEINS_full | 1113 | 2 | 39.06 | 72.82 | 3.74 | 25 | 1 |
| SYNTHETICnew | 300 | 2 | 100.00 | 196.25 | 3.93 | 9 | 1 |
| Synthie | 400 | 4 | 95.00 | 172.93 | 3.62 | 20 | 15 |
| ogbg-molpcba | 437,929 | 128 | 25.96 | 28.10 | 2.16 | 4 | 9 |

## B HYPERPARAMETERS USED FOR MODEL SELECTION IN THE GRAPH CLASSIFICATION TASK

For some kernels, only a subset of the hyperparameters was optimized, while the rest of the hyperparameters were kept fixed.

Table 6: Hyperparameters used for model selection in the graph classification experiments .

| Model | Layers | Convs per layer | Batch size | Learning rate | Hidden units | Epochs | L2 | Dropout | Patience (loss, acc) | Optimizer | Scheduler | Dense dim | Embed. dim | Neighbors Aggregation |
|---|---|---|---|---|---|---|---|---|---|---|---|---|---|---|
| DGNN | 2,3,4 | 1 | 16 | 1e−4 | 32, 64 | 1000 | – | 0.5 | 500, 500 | Adam | – | 128 | – | mean, max, sum |
| GIN | see hidden units | 1 | 32, 128 | 1e−2 | 32 (5 layers), 64 (5 layers), 64 (2 layers), 32 (3 layers) | 1000 | – | 0, 0.5 | 500, 500 | Adam | StepLR (step 50, $\gamma$=0.5) | – | – | sum |
| GraphSAGE | 3, 5 | 1 | 16 | 1e−2, 1e−3, 1e−4 | 32, 64 | 1000 | – | 0 | 500, 500 | Adam | – | – | – | mean, max, sum |
| InfoGraph | 3, 5 | – | 16, 32 | 1e−2, 1e−3 | 32, 64, 128 | 100 | 0, 1e−4 | 0, 0.1, 0.3 | 500, 500 | Adam | ReduceLROnPlateau ($\gamma$=0.5) | – | – | sum |
| GNN | 3, 5 | 1 | 16 | 1e−2, 1e−3, 1e−4 | 32, 64 | 1000 | – | 0, 0.5 | 500, 500 | Adam | StepLR (step 50, $\gamma$=0.5) | – | – | mean, max, sum |
| GraphCL | 2, 3 | 1 | 32 | 1e+3 | 64, 128 | 200 | – | 0 | 500, 500 | Adam | – | – | – | mean, max, sum |
| FNP | 2, 3, 4 | 1 | 32 | 1e−3 | 32, 64 | 1000 | – | 0.0, 0.2, 0.5 | 500, 500 | Adam | – | – | – | sum |
| PNA | 2, 3, 2 | 1 | 32 | 1e−3 | 32, 64 | 1000 | – | 0 | 500, 500 | Adam | – | – | – | max |
| PDF | 2, 3 | 1 | 32 | 5e−4 | 64, 129 | 300 | 1e−2 | 0 | 500, 500 | Adam | StepLR (step 50, $\gamma$=0.5) | – | – | mean |

Table 7: Hyperparameters used in the kernels for model selection in the graph classification task.

| Kernel | Fixed | Validation-tuned |
|---|---|---|
| SM | $k = 3$ | – |
| SP | – | – |
| ML | $\gamma = 0.01$, $\eta = 0.01$, $\hat{p} = 10$ | $l_{\max} \in \{0, \ldots, 5\}$, $\tilde{c} \in \{50, 100, 200, 300\}$ |
| PK | $w = 10^{-5}$ | $T \in \{1, \ldots, 6\}$ |
| HSPPK-WL | Iterations = 20 (100 for SYNTHIE) | $h \in \{0, \ldots, 5\}$ |
| HSPPK-SP | Iterations = 20 (100 for SYNTHIE) | $h \in \{0, \ldots, 5\}$ |
| GH | – | Linear kernel / Gaussian kernel |
| linearFGW-RAW | RBF kernel, $\gamma = 0.1$ | $\alpha \in \{0.1, 0.5, 0.9\}$, GWB layers = 5, OT layers $\in \{3, 5\}$, Iter $\in \{1, 2, 3\}$, $\gamma_{\text{kernel}} \in \{0.01, 0.1, 1.0\}$ |
| linearFGW-WL1 | RBF kernel, $\gamma = 0.1$ | same as above |
| linearFGW-WL2 | RBF kernel, $\gamma = 0.1$ | same as above |
| WL (disc) | – | Iterations $\in \{0, \ldots, 5\}$ |
| WLOA | – | Iterations $\in \{0, \ldots, 5\}$ |
| WWL | – | Iterations $\in \{0, \ldots, 7\}$, Sinkhorn $\in \{\text{False}, \text{True}\}$, $\gamma \in \{0.01, 0.1, 1, 10\}$ |
| NP | – | Iterations $\in \{0, \ldots, 5\}$, Linear / Gaussian kernel |
| NSPDK | $D = 1$, $R = 4$ | – |
| NSPDK (disc) | $D = 1$, $R = 4$ | – |
| NSPPK | $D = 1$, $R = 4$, $R' = 1$, threshold $t = 8$, nbits $n = 16$ | |

## B.1 ROBUSTNESS STUDY HYPERPARAMETERS

Table 8 summarizes the configurations used for the diagonal dominance / robustness experiments (Section 5.2). Neural baselines (GIN-Random, GraphCL, InfoGraph) were run with a common lightweight setup , while classical kernels (GraphHopper, Propagation Kernel) followed their standard definitions. NSPPK used the same fixed configuration as in the main experiments.

Table 8: Hyperparameters for robustness / diagonal dominance analysis.

| Method | Configuration |
|---|---|
| **NSPPK** | $R = 1$, $D = 4$, $R' = 1$, $t = 8$, $n_{\text{bits}} = 12$ |
| **GIN-Random** | 3 layers GIN, hidden dim=32, MLP layers=2, pooling=sum, epochs=200, lr=0.01, seed=42, orthogonal init, no supervision |
| **GraphCL** | Same GIN backbone as above, contrastive pretraining with augmentations, 200 epochs, lr=0.01 |
| **InfoGraph** | Same GIN backbone as above, maximizing mutual information, 200 epochs, lr=0.01 |
| **GraphHopper** | Shortest-path kernel, weight decay $w = 10^{-5}$, $t_{\max} \in \{1, 2, 3, 4, 5\}$ |
| **Propagation Kernel** | Attribute propagation with $M = L1$ distance, 5 iterations |

For Infograph,GraphCl and Gin-Random, we generate a dataset of 50000 graphs similar to G but the number of nodes was set to range from 50 to 250(as for the model to be able to detect node dropping).

## C NSPPK RELATION TO NSPDK

In this section we formalise the separation between NSPPK and NSPDK under a realistic, bounded-radius regime. Throughout, we adopt an *idealised hashing model*: each finite rooted (labelled) subgraph is mapped to a unique identifier that is invariant under isomorphism and never collides for

non-isomorphic subgraphs. This allows us to reason directly at the level of isomorphism types; the practical finite-bit SHA-based implementation only approximates this assumption.

*(NSPPK strictly dominates NSPDK under a fixed anchor radius).* There exists an infinite family of pairs of non-isomorphic, discretely labelled graphs $(G_n, H_n)$ and a constant anchor radius $R_0 \in \mathbb{N}$, $R_0 \geq 1$, such that:

1. For every $n$ and every choice of NSPDK parameters $(R, D)$ with $0 \leq R \leq R_0$, the NSPDK feature vectors of $G_n$ and $H_n$ are identical.

2. For every $n$ there exists a connector radius $R' \geq 0$ such that NSPPK with anchor radius $R \leq R_0$ and connector radius at most $R'$ assigns *distinct* feature vectors to $G_n$ and $H_n$.

Hence, under a fixed anchor-radius budget, NSPPK is strictly more expressive than NSPDK.

We give an explicit separating construction. In the construction below one can take $R_0 = 1$, so the above statement holds for some finite $R_0$ that does not grow with $n$.

**Construction.** For $n \geq 6$, let $G_n$ and $H_n$ be graphs obtained from a cycle $C_n$ by adding edges only between nodes lying on shortest paths between a fixed pair of antipodal vertices $(u, v)$. In $G_n$, add edges so that the union of all shortest $u{\leftrightarrow}v$ paths induces a single cycle. In $H_n$, add edges so that the union of all shortest $u{\leftrightarrow}v$ paths induces two internally node-disjoint shortest paths. All nodes carry the same discrete label.

**Step 1: NSPDK collapses $(G_n, H_n)$ for all $R \leq R_0$.** For this construction, fix $R_0 = 1$. By construction, for every $n$ and every vertex $x$, the $r$-hop neighborhoods $N_r(x)$ in $G_n$ and $H_n$ are isomorphic for all $0 \leq r \leq R_0$, and the pairwise graph distances $d(x, y)$ coincide for every vertex pair $(x, y)$. Therefore, for any NSPDK parameter choice $(R, D)$ with $R \leq R_0$, each feature of the form

$$\big(N_r(x),\, d(x, y),\, N_r(y)\big), \qquad 0 \leq r \leq R \leq R_0,$$

occurs with identical multiplicity in both graphs. Under idealised isomorphism-aware hashing, the NSPDK feature vectors for $G_n$ and $H_n$ are therefore identical for all $(R, D)$ with $R \leq R_0$. This proves the first item with a concrete finite choice of $R_0$.

**Step 2: NSPPK distinguishes $(G_n, H_n)$ with local radii.** Consider the anchor pair $(u, v)$. The unions of all shortest $u{\leftrightarrow}v$ paths,

$$U_{G_n}(u, v) \quad \text{and} \quad U_{H_n}(u, v),$$

are non-isomorphic: in $G_n$ this union induces a cycle, whereas in $H_n$ it induces two parallel paths. Hence, for every connector radius $r' \geq 0$, the connector subgraphs

$$C_{r'}^{G_n}(u, v) = N_{r'}\big(U_{G_n}(u, v)\big), \qquad C_{r'}^{H_n}(u, v) = N_{r'}\big(U_{H_n}(u, v)\big)$$

are non-isomorphic.

NSPPK includes features of the form

$$\big(N_r(u),\, C_{r'}(u, v),\, N_r(v)\big), \qquad 0 \leq r \leq R \leq R_0.$$

Since $N_r(u)$ and $N_r(v)$ are identical across $(G_n, H_n)$ for all $r \leq R_0$, the non-isomorphism of the connector subgraphs $C_{r'}(u, v)$ forces distinct feature identifiers under idealised hashing, already for $r' = 0$ (i.e., using the bare union of shortest paths). Thus NSPPK separates $(G_n, H_n)$ even when restricted to the same finite anchor-radius budget $R_0$.

**Remark (large-radius NSPDK).** If $R$ were allowed to grow with $n$ so that $R \geq \mathrm{diam}(G_n)$, then $N_R(x)$ would equal the entire graph, enabling NSPDK to distinguish the pair under idealised hashing. The proposition therefore compares NSPDK and NSPPK under a fixed, finite anchor-radius budget $R_0$ that does not scale with the graph size $n$, which is the practically relevant regime.

# D  LARGE SCALE EMBEDDING EXPERIMENT: QM9

Figure 7: NSPPK vectorization time for QM9 dataset as a function of number of bits.

Figure 7 illustrates the time required for NSPPK to vectorize the QM9 Blum & Reymond (2009) dataset as a function of the number of bits hyperparameter (nbits). As the number of bits increases, the vectorization time rises accordingly, though the rate of increase is not uniform. Up to 11 bits, the computation time remains within a small range (under 7 minutes), demonstrating NSPPK's efficiency in handling large-scale datasets.

However, a sharp increase in computation time occurs from 12 to 14 bits due to memory swapping, where the system resorts to using slower secondary storage instead of RAM. This significantly degrades performance, further emphasizing the importance of efficient memory usage when handling high-bit representations in large-scale datasets.

At 15 bits, the vectorization process fails due to excessive memory allocation requirements. This is a consequence of the exponential growth of the feature space: 15 bits corresponds to a $2^{15}$-dimensional representation per graph, resulting in an immense memory footprint when applied to over 129,000 molecular graphs. While this represents a practical upper bound for single-machine processing, it highlights the need for optimized memory management strategies for ultra-high-dimensional embeddings.

Despite this limitation, NSPPK remains an effective and scalable approach for graph learning tasks, provided that memory usage is carefully managed when selecting the number of bits. Additionally, potential optimizations such as sparse representations, dimensionality reduction, or distributed processing could further enhance its applicability to even larger datasets.

The QM9 dataset itself consists of over 129,000 molecular graphs with 16 continuous node attributes, making it a computationally intensive benchmark. The results confirm that NSPPK successfully processes datasets of this magnitude while maintaining practical computation times, reinforcing its utility for real-world graph-based applications.

# E  DENSITY-SENSITIVITY EXPERIMENT

To explicitly study the impact of graph density on runtime, we conducted a controlled *density-sensitivity experiment* on synthetic Erdős–Rényi graphs. This experiment complements our theoretical complexity analysis by measuring how NSPPK feature extraction scales as the average node degree increases.

**Experimental setup.** We generated graphs according to the Erdős–Rényi model $G(n, p)$ with a fixed number of nodes $n = 300$ and varying expected average degree $k$. For each value of $k$, the edge probability was set to $p = k/(n - 1)$. All graphs were assigned trivial, identical labels and no node attributes so as to isolate the effect of graph density. We measured the time required to extract NSPPK features using our standard configuration ($R = 1, D = 4, R' = 1, nbits = 11$) For each $k$, a single graph instance was generated and timed.

**Results.** Table 9 reports NSPPK feature extraction time as a function of the average degree.

Table 9: NSPPK feature extraction time vs. average graph degree on Erdős–Rényi graphs ($n = 300$).

| Avg. Degree | Time (s) |
|---|---|
| 2 | 2.29 |
| 27 | 66.11 |
| 52 | 102.40 |
| 77 | 150.75 |
| 102 | 202.39 |
| 127 | 256.25 |
| 152 | 308.38 |
| 177 | 355.59 |
| 202 | 388.98 |
| 227 | 412.80 |
| 252 | 425.54 |
| 277 | 429.48 |

**Discussion.** As the average degree increases from 2 to 277, the feature extraction time grows from approximately $2.3\,\mathrm{s}$ to $430\,\mathrm{s}$. The observed trend is close to linear in the number of edges $|E|$, which is consistent with the theoretical $O(|E|)$ scaling of the NSPPK hashing procedure for fixed radii and degree cutoff. Notably, even in the near-complete regime (average degree $\approx 277$ out of a maximum of 299), NSPPK remains computationally practical, requiring only a few minutes to process a dense 300-node graph.

# F ADDITIONAL EXPERIMENTS ON CITATION NETWORKS

To assess the robustness of NSPPK beyond molecular graphs, on node classification tasks, we additionally evaluated it on three widely used citation network benchmarks with fundamentally different structural properties: **Cora**, **CiteSeer**, and **PubMed**. These graphs exhibit higher-degree outliers, broader graph diameters, and citation-style connectivity patterns, in contrast to the small and sparse molecular graphs considered elsewhere in this paper.

Table 10: NSPPK performance on citation networks (80/20 split, seed 42).

| Dataset | #Nodes | #Edges | Avg Deg. | Max Deg. | Encoding Time (s) |
|---|---|---|---|---|---|
| Cora | 2,708 | 5,278 | 3.90 | 168 | 0.7897 |
| Citeseer | 3,327 | 4,552 | 2.74 | 99 | 0.7447 |
| PubMed | 19,717 | 44,324 | 4.50 | 171 | 0.8747 |

We compared NSPPK against a set of commonly used graph neural network architectures for node classification: GCN (Kipf & Welling, 2017a), GAT (Veličković et al., 2018), GraphSAGE (Hamilton et al., 2017), GIN (Xu et al., 2019b), SGC (Wu et al., 2019), APPNP (Klicpera et al., 2019), and a feature-only MLP.

All GNN models were trained directly on the original citation graphs using the same 80/20 train–test split as NSPPK. For GCN, GIN, GraphSAGE, and GAT, we used two-layer architectures with a hidden dimension of 64, ReLU activations, and dropout applied between layers. GAT employed eight attention heads in the first layer. APPNP was implemented as a two-layer MLP followed by personalized PageRank propagation with $K=10$ steps and teleport probability $\alpha=0.1$. SGC used a single linear classifier with $K=2$ propagation steps. The MLP baseline consisted of a two-layer feed-forward network operating solely on node features, without using graph structure.

All models were trained using the Adam optimizer (learning rate 0.01, weight decay $5 \times 10^{-4}$) for 200 epochs. Model selection relied solely on the fixed training split, without tuning hyperparameters per dataset. Performance is reported on the held-out test nodes in terms of macro-averaged ROC–AUC (one-vs-rest).

For NSPPK, followed a standard node classification setup with an 80/20 train–test split and a fixed random seed of 42. Crucially, we used *exactly the same NSPPK hyperparameters* as for all molecular datasets ($R = 1$, $D = 4$, $R' = 0$, 11-bit hashing), without any dataset-specific tuning. Continuous node attributes were projected to 24 dimensions using SVD. Discrete node labels required by NSPPK were obtained by running $k$-means clustering on the node attributes with $k = 5$, assigning each node its cluster membership as a discrete label.

Table 11: Test ROC–AUC (macro, one-vs-rest) on citation networks with an 80/20 node classification split.

| Dataset | GraphSAGE | GAT | APPNP | GCN | SGC | NSPPK+XGB | MLP | GIN |
|---|---|---|---|---|---|---|---|---|
| CiteSeer | 0.9422 | 0.9454 | 0.9310 | 0.9257 | 0.9432 | 0.9253 | 0.9247 | 0.9084 |
| Cora | 0.9846 | 0.9874 | 0.9884 | 0.9840 | 0.9874 | 0.9636 | 0.9502 | 0.9669 |
| PubMed | 0.9759 | 0.9615 | 0.9622 | 0.9659 | 0.9396 | 0.9686 | 0.9714 | 0.9663 |

Overall, these results demonstrate that NSPPK generalizes well to non-molecular domains and heterogeneous graph topologies, while retaining its deterministic nature and predictable runtime under a single, fixed hyperparameter configuration.

# G  SCALABILITY OF NSPPK VECTORIZATION

To evaluate NSPPK's scalability, we vectorized the **QM9** dataset (~112,000 graphs) using a fixed configuration: **12-bit hash size**, maximum **radius = 1**, **distance = 4**, and **connector path = 1**. This setup balances expressiveness and efficiency, making it suitable for large-scale benchmarks. As shown in Figure 8, the total vectorization time decreases almost linearly with the number of CPU cores, completing in under 10 minutes on 112 cores. This confirms NSPPK's efficient parallelization and practicality for large datasets.

Figure 8 presents the results on a log-log scale. The x-axis indicates the number of CPU cores, and the y-axis shows the total vectorization time. The observed trend is close to ideal linear scaling: doubling the number of cores results in approximately half the runtime. This demonstrates that NSPPK's feature extraction process incurs minimal synchronization or coordination overhead.

Experiments were conducted on a dual-socket Intel server equipped with 2× Intel Xeon Gold 6330 CPUs @ 2.00 GHz, each providing 28 physical cores (56 threads), for a total of 112 logical CPU cores. The machine had 2 NUMA nodes, 70 MB of shared L2 cache, and 84 MB of L3 cache. Despite relying solely on CPU resources, NSPPK scaled efficiently across all available cores. For example, complete vectorization of the QM9 dataset was achieved in under 10 minutes, demonstrating the method's practicality for real-world, large-scale deployment.

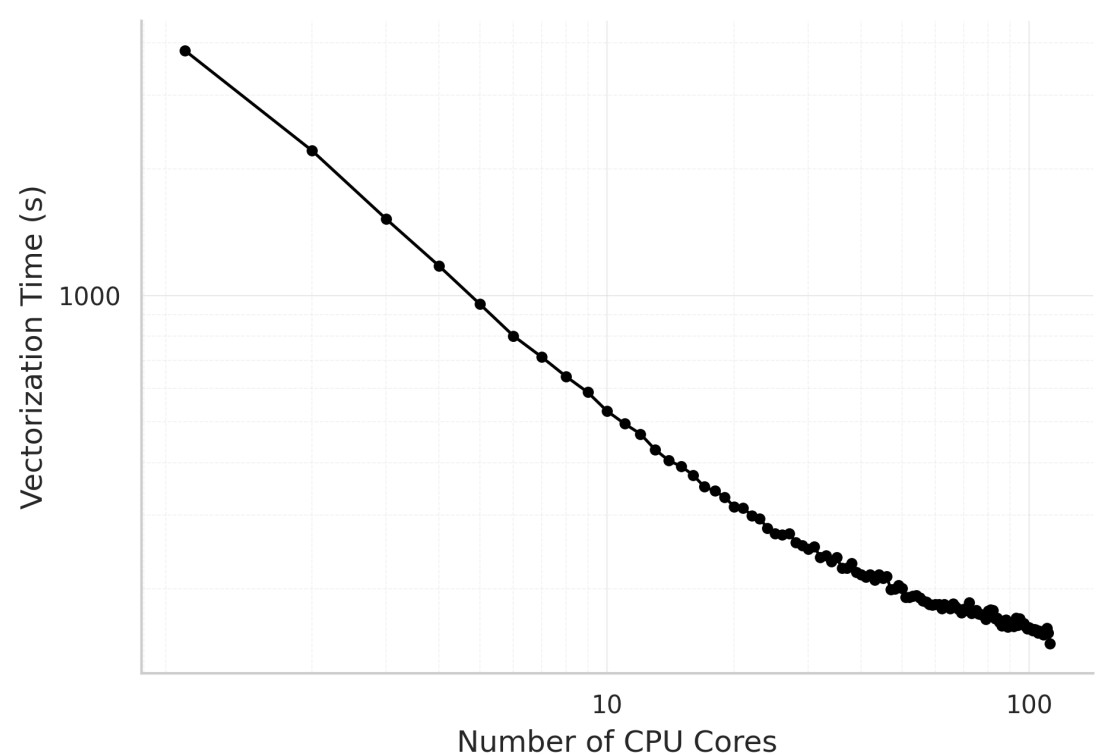

Figure 8: Vectorization time vs. number of CPU cores on QM9 (log-log scale). NSPPK demonstrates excellent parallel scalability, reducing total vectorization time from over an hour (using a single core) to under 10 minutes with 112 CPU cores. This shows near-linear performance gains with increased parallelization.

# H   FURTHER VISUALIZATION: ACCURACY–TIME TRADE-OFF

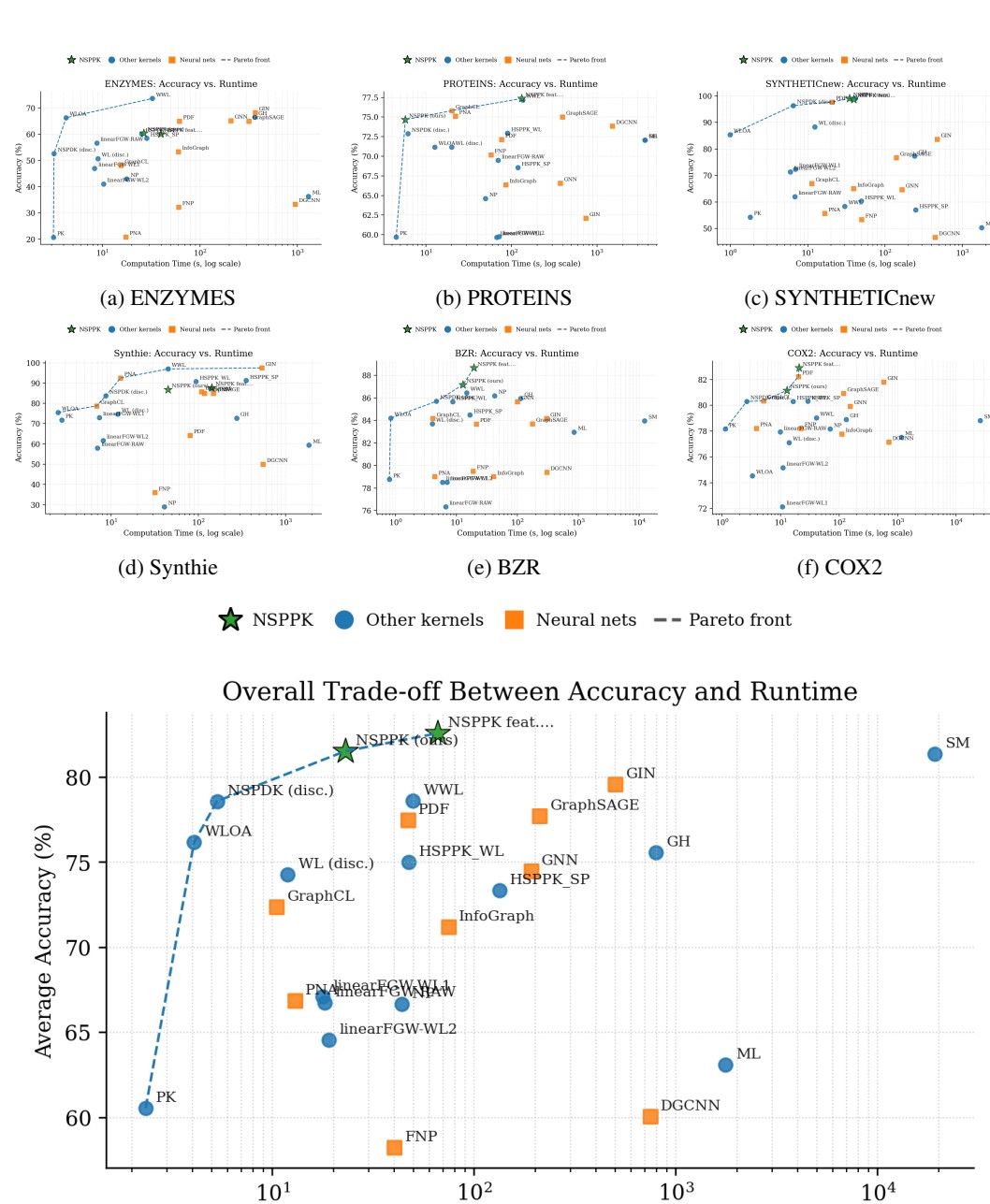

(a) ENZYMES

(b) PROTEINS

(c) SYNTHETICnew

(d) Synthie

(e) BZR

(f) COX2

(g) Overall (average across datasets)

Figure 9: **Accuracy–time trade-off (log time).** Markers: green star = NSPPK, blue circle = other kernels, orange square = neural nets. Dashed line = Pareto front.

**Summary.**   Figure 9 shows that, for most datasets, NSPPK (green star) is on or close to the Pareto front. Methods spread along the efficiency–accuracy spectrum: several are faster but less accurate, while others gain accuracy at a higher computational cost. On the aggregated panel, NSPPK remains on the global frontier, indicating a favorable accuracy–time balance overall.

## I  RUNTIME RESULTS FOR THE SMALL-SCALE DATASETS EXPERIMENTS REPORTED IN THE MAIN PAPER

Table 12: Runtime (seconds) **with** node attributes (**lower is better**).

| Method | SYNTH | SYNTHIE | BZR | COX2 | ENZ | PROT | Avg | Rank |
|---|---|---|---|---|---|---|---|---|
| SM | **TIMEOUT** | **TIMEOUT** | 12274.00 s | 25927.96 s | **TIMEOUT** | **TIMEOUT** | 190100.98 s | 16.00 |
| SP | **TIMEOUT** | **TIMEOUT** | **TIMEOUT** | **TIMEOUT** | **TIMEOUT** | **TIMEOUT** | – | – |
| ML | 1777.93 s | 1848.90 s | 849.20 s | 1155.49 s | 1295.99 s | 3628.70 s | 1759.70 s | 13.83 |
| PK | 1.81 s | 2.73 s | 0.79 s | 1.13 s | 3.14 s | 4.48 s | 2.51 s | 2.33 |
| HSPPK_WL | 49.54 s | 93.86 s | 8.63 s | 16.38 s | 25.20 s | 89.90 s | 47.59 s | 10.67 |
| HSPPK_SP | 249.75 s | 353.70 s | 16.52 s | 29.22 s | 28.35 s | 119.32 s | 132.81 s | 12.67 |
| GH | 242.05 s | 274.63 s | 112.43 s | 132.63 s | 365.25 s | 3647.21 s | 1129.03 s | 14.67 |
| NP | 41.57 s | 40.84 s | 42.32 s | 69.80 s | 17.69 s | 49.58 s | 43.63 s | 9.67 |
| linearFGW-RAW | 6.82 s | 7.00 s | 6.63 s | 9.84 s | 8.73 s | 69.67 s | 18.11 s | 6.67 |
| linearFGW-WL1 | 6.94 s | 7.33 s | 5.89 s | 10.75 s | 8.29 s | 67.19 s | 17.07 s | 6.00 |
| linearFGW-WL2 | 5.95 s | 8.09 s | 7.02 s | 10.92 s | 10.26 s | 71.63 s | 18.98 s | 6.33 |
| WL (disc.) | 12.33 s | 12.00 s | 4.00 s | 14.00 s | 9.00 s | 20.00 s | 11.22 s | **1.67** |
| WLOA | 0.99 s | 2.47 s | 0.83 s | 3.27 s | 4.24 s | 12.67 s | 4.41 s | 3.17 |
| WWL | 29.96 s | 45.00 s | 14.71 s | 40.89 s | 32.54 s | 134.95 s | 49.84 s | 11.67 |
| NSPDK (disc.) | 6.50 s | 8.71 s | 4.69 s | 2.64 s | 3.18 s | 6.13 s | 5.64 s | 3.17 |
| **NSPPK (ours)** | 34.81 s | 44.97 s | 12.75 s | 12.75 s | 26.45 s | 5.73 s | 22.41 s | 6.33 |

Table 13: Neural runtimes (seconds) **with** node attributes (**lower is better**).

| Method | SYNTH | SYNTHIE | BZR | COX2 | ENZ | PROT | Avg | Rank |
|---|---|---|---|---|---|---|---|---|
| DGCNN | 443.59 s | 551.36 s | 304.88 s | 704.74 s | 948.88 s | 1512.29 s | 744.29 s | 9.83 |
| GraphSAGE | 139.94 s | 117.92 s | 175.30 s | 118.97 s | 317.84 s | 394.02 s | 210.66 s | 7.33 |
| InfoGraph | 39.25 s | 109.14 s | 40.40 s | 111.40 s | 59.95 s | 85.72 s | 74.31 s | 5.00 |
| GIN | 474.24 s | 535.34 s | 302.44 s | 579.86 s | 369.06 s | 742.46 s | 500.57 s | 9.17 |
| GraphCL | 11.33 s | 6.83 s | 4.10 s | 5.11 s | 15.51 s | 20.04 s | 10.49 s | **1.17** |
| GNN | 165.92 s | 148.83 s | 100.28 s | 153.89 s | 207.18 s | 371.30 s | 191.23 s | 7.50 |
| FNP | 49.97 s | 31.88 s | 18.72 s | 22.23 s | 60.51 s | 57.51 s | 40.14 s | 4.17 |
| PNA | 16.67 s | 12.87 s | 4.38 s | 3.86 s | 17.33 s | 22.14 s | 12.88 s | 1.83 |
| PDF | 20.59 s | 80.49 s | 21.10 s | 20.13 s | 61.43 s | 76.70 s | 46.74 s | 4.17 |
| **NSPPK feat. (XGB)** | 39.35 s | 142.70 s | 19.38 s | 20.62 s | 39.54 s | 131.96 s | 65.59 s | 4.83 |

## J  ADDITIONAL RESULTS: NO-ATTRIBUTE SETTING

Tables 14 and 15 report kernel and neural network accuracy, respectively, when node attributes are removed. This isolates the structural contribution of the methods. We observe that NSPPK maintains strong relative performance even without attributes, underscoring its robustness.

Table 14: Classification accuracy (%) **without** node attributes (-) with Avg Rank.

| Method | SYNTHETIC new | Synthie | BZR | COX2 | ENZYMES | PROTEINS | Avg Rank |
|---|---|---|---|---|---|---|---|
| SM | **TIMEOUT** | **TIMEOUT** | $79.02 \pm 1.10$ | $78.16 \pm 0.81$ | **TIMEOUT** | **TIMEOUT** | 8.75 |
| SP | **TIMEOUT** | **TIMEOUT** | **TIMEOUT** | **TIMEOUT** | **TIMEOUT** | **TIMEOUT** | – |
| ML | $67.22 \pm 9.51$ | $58.00 \pm 13.85$ | $86.63 \pm 3.81$ | $77.95 \pm 4.63$ | $33.66 \pm 5.31$ | $72.80 \pm 3.80$ | **4.33** |
| PK | $61.33 \pm 7.33$ | $38.75 \pm 7.60$ | $78.77 \pm 1.01$ | $78.16 \pm 8.07$ | $18.33 \pm 5.22$ | $58.48 \pm 4.29$ | 11.17 |
| HSPPK_WL | $50.00 \pm 0.00$ | $54.25 \pm 1.14$ | $80.26 \pm 3.02$ | $72.41 \pm 17.06$ | $16.50 \pm 2.73$ | $63.80 \pm 5.76$ | 11.92 |
| HSPPK_SP | $58.00 \pm 7.18$ | $47.75 \pm 6.75$ | $76.26 \pm 9.39$ | $77.31 \pm 4.70$ | $21.00 \pm 5.92$ | $46.20 \pm 4.70$ | 12.17 |
| GH | $59.33 \pm 9.28$ | $52.25 \pm 4.10$ | $81.25 \pm 2.40$ | $77.30 \pm 3.14$ | $25.17 \pm 3.98$ | $71.61 \pm 4.32$ | 7.33 |
| NP | $97.00 \pm 3.15$ | $47.60 \pm 0.00$ | $84.16 \pm 5.65$ | $80.29 \pm 3.42$ | $36.70 \pm 0.53$ | $69.99 \pm 3.88$ | 5.50 |
| linearFGW-RAW | $57.33 \pm 6.29$ | $44.75 \pm 8.91$ | $80.52 \pm 3.76$ | $76.24 \pm 4.74$ | $23.00 \pm 4.88$ | $71.35 \pm 4.56$ | 10.33 |
| linearFGW-WL1 | $56.00 \pm 11.72$ | $54.50 \pm 8.28$ | $80.99 \pm 5.24$ | $76.45 \pm 1.86$ | $24.50 \pm 4.78$ | $70.17 \pm 4.81$ | 9.00 |
| linearFGW-WL2 | $48.67 \pm 7.33$ | $51.75 \pm 4.19$ | $79.48 \pm 5.48$ | $74.74 \pm 5.12$ | $22.33 \pm 5.59$ | $69.54 \pm 3.12$ | 11.67 |
| WL (disc.) | $79.00 \pm 12.39$ | $54.75 \pm 3.94$ | $87.90 \pm 3.92$ | $78.17 \pm 3.48$ | $40.17 \pm 7.54$ | $69.00 \pm 4.08$ | **4.00** |
| WLOA | $81.00 \pm 6.16$ | $50.75 \pm 4.62$ | $83.71 \pm 8.36$ | $78.16 \pm 2.75$ | $42.67 \pm 4.84$ | $74.49 \pm 3.53$ | 4.75 |
| WWL | $50.00 \pm 0.00$ | $27.50 \pm 0.00$ | $78.77 \pm 1.01$ | $78.16 \pm 0.80$ | $16.67 \pm 0.00$ | $55.57 \pm 0.17$ | 12.58 |
| NSPDK | $95.33 \pm 3.72$ | $51.25 \pm 5.01$ | $85.68 \pm 4.04$ | $77.09 \pm 3.84$ | $35.67 \pm 9.22$ | $71.33 \pm 3.06$ | 6.50 |
| NSPDK (disc.) | $95.33 \pm 3.72$ | $51.25 \pm 5.01$ | $85.68 \pm 4.04$ | $77.09 \pm 3.84$ | $35.67 \pm 9.22$ | $71.33 \pm 3.06$ | 6.50 |
| **NSPPK (ours)** | $98.00 \pm 3.93$ | $53.00 \pm 4.30$ | $87.65 \pm 4.54$ | $77.73 \pm 3.96$ | $33.17 \pm 5.80$ | $71.34 \pm 4.06$ | 4.67 |

*Note:* Avg Rank averaged over available cells; lower is better. TIMEOUT/N/A omitted per dataset.

Table 15: Neural networks: classification accuracy (%) **without** node attributes (-) and Avg Rank.

| Method | SYNTHETIC new | Synthie | BZR | COX2 | ENZYMES | PROTEINS | Avg Rank |
|--------|---------------|---------|-----|------|---------|----------|----------|
| DGCNN | $44.67 \pm 6.86$ | $25.25 \pm 8.69$ | $81.98 \pm 2.20$ | $78.22 \pm 0.07$ | $26.80 \pm 7.09$ | $73.22 \pm 3.48$ | 6.83 |
| GraphSAGE | $43.33 \pm 5.58$ | $33.00 \pm 8.20$ | $83.70 \pm 5.59$ | $80.30 \pm 0.03$ | $48.17 \pm 7.58$ | $74.93 \pm 2.82$ | 4.92 |
| InfoGraph | $67.33 \pm 21.54$ | $42.75 \pm 13.53$ | $75.05 \pm 15.04$ | $69.02 \pm 0.20$ | $53.33 \pm 4.79$ | $63.07 \pm 4.69$ | 6.17 |
| GIN | $53.00 \pm 9.71$ | $43.25 \pm 12.53$ | $73.95 \pm 3.30$ | $79.91 \pm 0.08$ | $42.67 \pm 7.68$ | $65.77 \pm 5.02$ | 6.17 |
| GraphCL | $50.00 \pm 8.69$ | $27.00 \pm 7.40$ | $79.99 \pm 3.62$ | $81.38 \pm 3.79$ | $37.50 \pm 5.12$ | $71.43 \pm 3.92$ | 6.08 |
| GNN | $43.33 \pm 5.58$ | $23.25 \pm 7.34$ | $84.66 \pm 4.60$ | $81.60 \pm 5.54$ | $48.50 \pm 5.80$ | $71.79 \pm 3.61$ | 5.42 |
| FNP | $50.00 \pm 8.69$ | $51.25 \pm 10.56$ | $81.73 \pm 3.52$ | $78.22 \pm 3.94$ | $35.83 \pm 7.79$ | $72.41 \pm 3.70$ | 5.33 |
| PNA | $46.00 \pm 7.72$ | $48.25 \pm 6.71$ | $78.76 \pm 4.33$ | $78.22 \pm 7.01$ | $18.83 \pm 7.07$ | $70.62 \pm 3.69$ | 7.33 |
| PDF | $50.00 \pm 8.69$ | $24.75 \pm 7.02$ | $84.43 \pm 4.63$ | $81.38 \pm 3.79$ | $52.00 \pm 5.26$ | $74.75 \pm 2.28$ | **4.08** |
| NSPPK feat. (XGBoost) | $91.00 \pm 4.73$ | $50.00 \pm 6.12$ | $89.60 \pm 3.29$ | $82.76 \pm 4.25$ | $41.83 \pm 5.55$ | $71.60 \pm 3.15$ | **2.83** |

## K ADDITIONAL RUNTIMES: NO-ATTRIBUTE SETTING

Tables 16 and 17 report computation times for kernels and neural networks without node attributes. While runtimes are generally shorter in this simplified setting, the relative ranking remains consistent: NSPPK achieves strong efficiency while preserving accuracy.

Table 16: Neural runtimes (seconds) **without** node attributes (**lower is better**).

| Method | SYNTH | SYNTHIE | BZR | COX2 | ENZ | PROT | Avg | Rank |
|--------|-------|---------|-----|------|-----|------|-----|------|
| DGCNN | 428.69 s | 562.06 s | 571.26 s | 237.85 s | 887.36 s | 1611.41 s | 716.10 s | 9.83 |
| GraphSAGE | 124.81 s | 161.54 s | 126.21 s | 140.69 s | 248.03 s | 395.38 s | 199.44 s | 7.33 |
| InfoGraph | 65.27 s | 104.54 s | 100.00 s | 87.92 s | 144.01 s | 241.47 s | 123.54 s | 5.00 |
| GIN | 388.95 s | 360.00 s | 358.57 s | 392.46 s | 440.30 s | 884.01 s | 470.38 s | 9.17 |
| GraphCL | 3.58 s | 2.09 s | 4.23 s | 5.42 s | 13.28 s | 17.61 s | 6.03 s | **1.17** |
| GNN | 81.93 s | 103.93 s | 28.95 s | 122.86 s | 247.42 s | 403.82 s | 164.49 s | 7.50 |
| FNP | 7.69 s | 22.24 s | 63.45 s | 15.82 s | 70.24 s | 38.84 s | 36.05 s | 4.17 |
| PNA | 3.32 s | 10.28 s | 3.83 s | 2.80 s | 8.59 s | 23.63 s | 8.41 s | 1.83 |
| PDF | 13.12 s | 25.58 s | 17.80 s | 23.45 s | 63.85 s | 67.34 s | 35.86 s | 4.17 |
| NSPPK feat. (XGB) | 32.91 s | 59.78 s | 4.28 s | 6.25 s | 7.43 s | 60.64 s | 28.38 s | 4.83 |

Table 17: Runtime (seconds) **without** node attributes (**lower is better**).

| Method | SYNTH | SYNTHIE | BZR | COX2 | ENZ | PROT | Avg | Rank |
|--------|-------|---------|-----|------|-----|------|-----|------|
| SM | TIMEOUT | TIMEOUT | 11853.30 s | 25478.50 s | TIMEOUT | TIMEOUT | 18665.90 s | 16.00 |
| SP | TIMEOUT | TIMEOUT | TIMEOUT | TIMEOUT | TIMEOUT | TIMEOUT | – | – |
| ML | 978.86 s | 1953.89 s | 158.40 s | 1509.62 s | 1792.43 s | 5661.68 s | 2009.15 s | 15.00 |
| PK | 0.59 s | 0.92 s | 0.23 s | 0.33 s | 0.78 s | 2.11 s | 0.83 s | 2.00 |
| HSPPK_WL | 38.74 s | 57.69 s | 8.76 s | 11.30 s | 19.83 s | 76.69 s | 35.50 s | 10.50 |
| HSPPK_SP | 285.63 s | 320.39 s | 18.33 s | 25.21 s | 31.37 s | 121.37 s | 133.72 s | 12.33 |
| GH | 178.47 s | 375.02 s | 99.82 s | 132.87 s | 365.25 s | 791.82 s | 323.21 s | 13.83 |
| NP | 66.93 s | 53.05 s | 44.90 s | 97.51 s | 69.69 s | 240.32 s | 95.07 s | 12.50 |
| linearFGW-RAW | 6.21 s | 8.21 s | 6.17 s | 8.75 s | 12.91 s | 76.17 s | 19.07 s | 7.83 |
| linearFGW-WL1 | 7.00 s | 8.44 s | 6.24 s | 10.22 s | 8.51 s | 52.96 s | 15.23 s | 7.83 |
| linearFGW-WL2 | 6.17 s | 7.53 s | 5.21 s | 11.27 s | 11.82 s | 74.40 s | 19.73 s | 7.17 |
| WL (disc.) | 0.14 s | 0.11 s | 0.05 s | 0.12 s | 0.13 s | 0.34 s | 0.15 s | **1.00** |
| WLOA | 0.96 s | 1.02 s | 0.95 s | 1.37 s | 3.32 s | 8.19 s | 2.63 s | 3.83 |
| WWL | 13.19 s | 23.60 s | 11.17 s | 29.40 s | 24.10 s | 90.97 s | 32.41 s | 10.83 |
| NSPDK | 5.78 s | 7.00 s | 3.07 s | 2.81 s | 2.24 s | 3.11 s | 3.67 s | 4.17 |
| NSPDK (disc.) | 5.78 s | 7.00 s | 3.07 s | 2.81 s | 2.24 s | 3.11 s | 3.67 s | 4.17 |
| **NSPPK (ours)** | 11.31 s | 13.33 s | 6.09 s | 6.09 s | 5.58 s | 4.72 s | 7.85 s | 7.00 |

## L LARGE-SCALE EXPERIMENT: MOLPCBA LEARNING CURVES AND EFFICIENCY

We evaluated NSPPK on the large-scale `ogbg-molpcba` benchmark from the Open Graph Benchmark suite Hu et al. (2020a), which includes 437,929 node attributed molecular graphs and 128 binary classification tasks. In practice, many of these tasks are both sparse (due to missing labels) and highly imbalanced.

## L.1 CASE STUDY: TARGET 95 FROM `OGBG-MOLPCBA`

We further examined **task 95**, which provides 48,853 positive examples, 293,968 negatives, and 95,108 molecules with missing labels. To analyze sample efficiency, we subsampled balanced datasets up to **78,164 labeled graphs** (positives and negatives in equal proportion) and varied the training set size from 100 to 250k examples. Each experiment was repeated with **five random seeds**, and average precision (AP) was reported. In parallel, we also trained each baseline once on the *full OGB scaffold split* (249,715 train, 29,826 validation, 29,427 test).

We compared NSPPK in combination with different downstream classifiers—logistic regression, random forest, and XGBoost—using both *sparse* and *dense* feature representations, against neural baselines including GIN, GAT, PNA, PDF, a generic GNN, and AttentiveFP (FNP). The distinction between sparse and dense refers only to feature storage: sparse matrices retain only nonzeros and are CPU-efficient, while dense mode expands full vectors (more memory, but occasionally favorable for GPU kernels).

For NSPPK, we fixed a single configuration ($R = 1$, $D = 4$, $R' = 1$, $n_{\text{bits}} = 16$) across all runs. (see Appendix L.2 for full implementation details of the graph neural netowks models used within this experiment).

Figure 10 summarizes the results. NSPPK shows strong sample efficiency, achieving higher AP than all neural baselines at small training sizes. Its runtime is also favorable: sparse variants in particular remain substantially faster to train than graph neural networks. At scale, PDF overtakes NSPPK in predictive performance, though the gap remains small. Interestingly, NSPPK combined with logistic regression can take as long as a GNN to train, but still delivers superior AP on small data regimes. Overall, NSPPK offers a simple, lightweight alternative that competes directly with neural methods. Figure 10 shows the resulting learning curves, with all NSPPK variants highlighted in red.

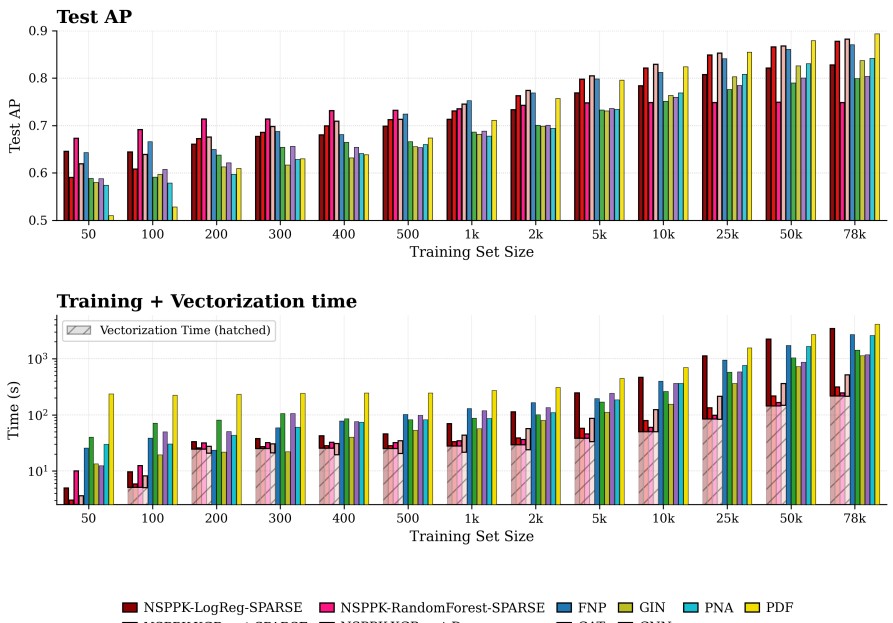

Figure 10: Learning curves on `ogbg-molpcba` (task 95). NSPPK (red) paired with different downstream classifiers is compared against neural baselines including GIN, GAT, PNA, PDF, and AttentiveFP (FNP).

## L.2 IMPLEMENTATION DETAILS FOR TASK 95 EXPERIMENTS

**NSPPK Configuration.** For all NSPPK experiments we fixed the parameters across classifiers:

$$R = 1, \quad D = 4, \quad R' = 1, \quad n_{\text{bits}} = 16.$$

Both sparse and dense representations were evaluated. Sparse mode stores only nonzero entries and is efficient on CPU-based models, while dense mode expands the full vectors, sometimes favorable for GPU-accelerated tree methods.

**Downstream Classifiers.** The exact settings for the classifiers paired with NSPPK features are given in Table 18.

Table 18: Classifiers used with NSPPK features on `ogbg-molpcba` (task 95).

| Classifier | Features | Configuration |
|---|---|---|
| Logistic Regression | Sparse | `saga`, max_iter=1000, $L_2$, $n_{\text{jobs}} = 64$ |
| Random Forest | Sparse | 500 trees, depth=5, $n_{\text{jobs}} = 64$, seed=42 |
| XGBoost | Dense | 1000 trees, depth=6, LR=0.03, subsample=0.8, colsample=0.8 |
| XGBoost | Sparse | Same as above, `hist` backend |

**Neural Baselines.** For comparison, we trained common GNN baselines with published hyperparameters. Table 19 summarizes their main configurations.

Table 19: Neural baselines and their configurations for task 95.

| Model | Main hyperparameters | Source |
|---|---|---|
| PNA | 2 layers, dim 64→32, batch 64/256, LR=0.001, Adam | Corso et al. (2020) |
| PDF (Basis-DGL) | 8 layers, dim 384, batch 64/256, LR=5e-4, AdamW | Yang et al. (2023b) |
| GAT | 2 layers, 64→32, 4/1 heads, LR=0.001, Adam | Veličković et al. (2018) |
| AttentiveFP (FNP) | 4 layers, dim 64, dropout=0.2, LR=0.001, Adam | Xiong et al. (2019) |
| GIN | 2 layers, dim 64→32, LR=0.001, Adam | Xu et al. (2019a) |
| GCN (OGB baseline) | 2 layers, dim 64→32, LR=0.001, Adam | Hu et al. (2020a) |

**Shared Training Setup.** All neural baselines were trained on the official OGB scaffold split (train: 249,715; validation: 29,826; test: 29,427). Loss: binary cross-entropy with logits (`BCEWithLogitsLoss`). Metrics: AP and ROC-AUC. Unless otherwise stated, all experiments were executed on CPU.

**Reproducibility.** Balanced-data experiments were repeated with five random seeds. Both feature extraction time and training time are reported in the main text.

