# OpenReview forum: "Scalable Graph Kernels with Continuous Attributes"
_ICLR.cc/2026/Conference — ICLR 2026 Conference Desk Rejected Submission_

### Official Review · Reviewer_49hc · 2025-10-25

**Soundness:** 3
**Presentation:** 2
**Contribution:** 2
**Rating:** 2
**Confidence:** 4

**Summary:**

This paper presents a graph kernel which generalizes the NSPDK kernel [1] and improves its expressive power. Unlike the NSPDK kernel, which considers only the distance between two nodes of a graph, the proposed kernel also takes into account the subgraph induced by the shortest paths connecting those two nodes. The kernel is evaluated on 6 small-scale and 1 large-scale graph classification dataset. The results indicate that the proposed kernel achieves high classification accuracies on the small-scale datasets and is competitive with the baselines.

[1] Costa, F., & De Grave, K. Fast neighborhood subgraph pairwise distance kernel. In ICML'10.

**Strengths:**

- The presented empirical results demonstrate that the proposed NSPPK kernel can achieve high classification accuracies on the node-attributed graph classification benchmark datasets, as it outperforms the other kernels on 2 out of the 6 considered datasets and the graph neural networks on 4 out of the 6 datasets.

- The example in Figure 1 serves as a motivation for the proposed kernel and demonstrates that it can produce features distinct from those of the NSPDK kernel from which it draws inspiration. Furthermore, the NSPPK kernel outperforms NSPDK in the empirical evaluation.

- The authors demonstrate that, due to the explicit feature maps that it produces, the kernel can be applied to very large datasets containing hundreds of thousands of samples.

**Weaknesses:**

- The presentation in Section 4 is unclear in several places. In particular, Subsection 4.3, which describes the features produced by the kernel, lacks clarity and would benefit from revision. It is difficult for the reader to understand what the features of $f_G^\theta$ correspond to. Also Subsection 4.5 needs to be revised since it is not clear how continuous features are exactly handled by the kernel.

- The authors show that the proposed kernel produces features which are not produced by the NSPDK kernel. However, I would suggest they also provide two non-isomorphic graphs which are both embedded to the same feature vector by NSPDK, but to different feature vectors by the proposed kernel.

- As discussed above, the proposed kernel outperforms the baseline kernels on 2 out of the 6 datasets. However, the improvements offered by the proposed kernel are only marginal (less that 1% absolute difference in average accuracy).

- The NSPPK can also compute explicit feature vectors for graphs whose nodes are assigned discrete labels. The paper would be strengthened if the authors also evaluated the kernel on benchmark datasets that contain such graphs (e.g., MUTAG, NCI1). As these datasets are well-studied, such an evaluation would clarify whether the kernel can compete with state-of-the-art approaches.

- The hyperparameters $R$ and $D$ cannot take large values since this will largely increase the computational complexity of the kernel. Furthermore, the complexity analysis presented in Subsection 4.6 seems incomplete since it does not consider the time to compute the neighborhood pair hash (which requires the rooted graph hash, etc.)

**Questions:**

Why aren't Tables 1 and 2 merged into a single Table? It is clear to me that the results of Table 1 are produced by the SVM classifier and those of Table 2 by XGBoost, but I would suggest the authors merge the two Tables and use two rows for the proposed kernel (one row for each classifier) or just report the results produced by SVM.

---

> ### Author Response · Authors · 2025-11-19
>
> $f_G^θ$ is the explicit feature vector of graph G under hyperparameters θ = (R, D, R'). Each coordinate corresponds to one hash bucket (i.e., one anchor–connector–anchor motif), and $f_G^θ[h]$ is simply the count of how many times that motif appears in G. Regarding subsection 4.5, the kernel handles continuous attributes in a simple way: every time a subgraph occurs in a graph, we know which nodes participated in it. We collect this information in a node–feature incidence matrix $F$. Then we aggregate the continuous attributes by multiplying the attribute matrix $A$ with $F$. In other words, for each feature with index $h$, we sum the attributes of all nodes that appear in subgraphs mapped to $h$. This produces an explicit vector in which each entry stores the aggregate of each continuous attribute present in that structural pattern.
>
>
>
> ##
> For the expressivity analysis see the answer to reviewer wKc1
>
>
> ##
> We appreciate the reviewer’s observation. NSPPK achieves the **best average rank** across all six datasets and performs consistently well rather than excelling on only one dataset and performing poorly on others. In contrast, several baseline kernels show higher variance: they may perform strongly on a specific dataset but drop substantially on others, leading to a worse overall rank. As shown in Figure. 7, NSPPK is consistently on the Pareto front of performance and runtime making it a good choice for low hyperparameter graph processing approach.
>
>
> ##
> Following the recommendation, we evaluated NSPPK on two standard discrete-label benchmark datasets, MUTAG and NCI1. For a fair comparison with NSPDK, we used the same hyperparameters for all runs: radius R = 1, distance D = 4, and connector radius R' = 1, with the same bit budget (11 bits) for both kernels.
> The results show that NSPPK is competitive and slightly improves over NSPDK on both datasets:
> MUTAG: NSPPK = 0.8775 ± 0.0479, NSPDK = 0.8561 ± 0.0487
> NCI1: NSPPK = 0.8574 ± 0.0227, NSPDK = 0.8460 ± 0.0096
>
> These experiments confirm that the proposed connector-based features also provide benefits in the purely discrete-label setting, without relying on continuous attributes.
>
>
>
> ##
>
> We fully agree that taking large values of the radii and distance parameters $(R,D,R')$ would  increase the computational cost. In our analysis we explicitly work in the regime where these are treated as small constants. Intuitively, larger distances could increase the number of nodes visited by the BFS, and larger radii would lead to large patterns, that would provide no additional discriminative power, since these patterns would not occur in more than a single instance.
> We also agree that the text should be clearer that the hashing cost is included in the complexity analysis: the bulk of the work is performed in the BFS up to depth $B = \max(R,D,R'),$ which visits at most $O(K^B)$ nodes with $K$ the maximum degree. Once this BFS tree is built, computing rooted graph hashes, union-of-shortest-path hashes, and neighborhood pair hashes only touches each node/edge in these BFS trees a constant number of times (a small fixed number of SHA256 and set/sequence hash calls). This adds an $ O(|V_{\text{BFS}}| + |E_{\text{BFS}}|)$
> term per node, which is asymptotically absorbed into the existing $O(|V| K^B)$ bound for fixed small $(R,D,R')$. We will update Subsection 4.6 to state this explicitly and make clear that the hashing cost does not change the asymptotic complexity, but only the constant factor.
>
> ##
>
> In our setting, the SVM experiments in Table 1 and the XGBoost experiments in Table 2 are not directly interchangeable, because they operate on *different representations*:
>
> - **Table 1 (SVM)** uses the **kernel Gram matrix**. This is the only scenario in which other graph kernels can be evaluated, since those do not produce explicit feature vectors.
> - **Table 2 (XGBoost)** uses the **explicit feature vectors** produced by our NSPPK encoder. Merging the two tables would therefore conflate these two evaluation regimes.

---

> > ### Comment · Reviewer_49hc · 2025-11-26
> >
> > I would like to thank the authors for their response. Most of my concerns have been addressed.
> >
> > From the pair of graphs provided in the response to reviewer wKc1, it is clear that NSPPK is more expressive than NSPDK. In fact, NSPPK can detect triangles whereas NSPDK cannot for the given values of $r$ and $d$. In addition, the reported test accuracy on NCI1 places NSPPK among the most effective approaches on this dataset (it even outperforms well-established GNNs).
> >
> > I am thus increasing my rating to 4. I still do not believe that the paper is ready for acceptance at a top-tier venue such as ICLR because of the following reasons:
> >
> > - I believe that the paper would greatly benefit from a major revision of section 4 to improve the clarity of the presentation.
> >
> > - The NSPPK kernel is significantly outperformed by at least one baseline on some datasets (e.g., ENZYMES). Therefore, a more thorough evaluation of the kernel (in terms of both performance and runtime) is needed. The experiments on the two standard graph classification datasets (MUTAG and NCI1) are in the right direction, but additional datasets should be included.
> >
> > I also do not agree that XGBoost and SVM use different representations. The primal form of the SVM could use explicit feature vectors (the ones that are fed to XGBoost) and is equivalent to the dual form, in the sense that they produce the same classifier. On the other hand, the GNNs that are included in Table 2 operate on a different graph representation than the one generated by NSPPK which is then passed to SVM or XGBoost.

---

> > > ### Author Response · Authors · 2025-11-30
> > >
> > > We thank the reviewer for the careful reassessment and for increasing the score. We  agree that Section 4 would benefit from improved clarity, and we have  revised it in order to make it easier to follow. We also acknowledge that NSPPK is not uniformly the top performer on every dataset; our objective is not to dominate each individual benchmark, but to demonstrate consistent and robust performance across datasets with continuous node attributes using a single fixed configuration. In this setting, NSPPK attains the best average rank among kernel methods and among all evaluated approaches, which we argue is a more meaningful indicator of generality than isolated peak performance.
> > > Graph classification benchmarks with continuous node attributes are relatively scarce,  we are using the classical ones using in the kernel literature(BZR, COX2, ENZYMES...). Since MUTAG and NCI1 lack such attributes, they do not directly evaluate NSPPK’s central contribution; we therefore emphasize attribute-rich datasets and report structure-only results for reference.
> > > While it is indeed possible to train a primal linear SVM ( LIBSVM) directly on NSPPK’s explicit feature vectors, this option is not available for the other kernel baselines, which are defined only implicitly through Gram matrices and do not expose explicit feature maps. As a result, a single unified table would necessarily mix two different evaluation regimes: (i) kernel-based SVMs for classical graph kernels and (ii) vector-based classifiers for NSPPK. We therefore separate the results to maintain a fair and transparent comparison: Table 1 compares NSPPK with other kernels under the common kernel-SVM setting, while Table 2 evaluates NSPPK as an explicit graph representation alongside GNN-based approaches.

---

### Official Review · Reviewer_iBo3 · 2025-10-31

**Soundness:** 3
**Presentation:** 3
**Contribution:** 2
**Rating:** 4
**Confidence:** 4

**Summary:**

The paper introduces the Neighborhood Subgraph Pairwise Path Kernel (NSPPK), a scalable and interpretable graph kernel designed for attributed graphs. NSPPK extends the NSPDK kernel by enriching the structural context through unions of shortest-path neighborhoods and directly integrating continuous node features without discretization. The method produces explicit, sparse graph-level embeddings that allow efficient similarity computation and scales near-linearly in the number of nodes and edges.

**Strengths:**

The paper presents a well-motivated extension of NSPDK that meaningfully increases expressiveness by incorporating shortest-path unions and continuous features.

NSPPK yields explicit sparse graph embeddings, making similarity evaluation efficient (single dot product) and enabling easy downstream application with classical classifiers.

**Weaknesses:**

Although the method is positioned as scalable, it is not compared against recent GNNs explicitly designed for large-scale settings (e.g., GraphSAINT, Cluster-GCN), nor against more recent graph transformer models such as GraphGPS. These baselines are directly relevant for claims regarding scalability and competitiveness.

Figure 4 shows that many NSPPK+classifier combinations either degrade in accuracy or do not significantly improve runtime relative to GNNs, except for the XGBoost pairing. A deeper discussion is needed to explain why certain classifier pairings behave poorly.

The “training-free” claim may be somewhat overstated, as classification performance still depends on training a downstream model (e.g., XGBoost, logistic regression). Clarifying this distinction would improve the positioning of the work.

**Questions:**

The experimental setup states that GPU clusters with A100s were used, which seems a lot given the model size is small. Could you clarify whether any part of NSPPK relies heavily on GPU acceleration, or if this was only for baselines?

---

> ### Author Response · Authors · 2025-11-19
>
> We agree that scalability comparisons are important. However, methods like GraphSAINT and Cluster-GCN are specifically designed for large-scale node-classification on a single massive graph, whereas our evaluation (following graph-kernel convention) focuses on graph-level prediction across many medium-sized graphs. Thus, applying GraphSAINT or Cluster-GCN directly is not meaningful for our setting.
> That said, to address the reviewer’s request, we additionally benchmarked NSPPK against a recent scalable graph transformer, GraphGPS, on a large graph-prediction task with increasing train set sizes. NSPPK is used as a training-free vectorizer followed by XGBoost, while GraphGPS requires full end-to-end training for each train-size.
>
> The results are shown below.
> | Train Size | GraphGPS Test AP | GraphGPS Total Time (s) | NSPPK+XGBoost Test AP | NSPPK+XGBoost Total Time (s) |
> |------------|------------------|---------------------------|------------------------|-------------------------------|
> | 50         | 0.543860         | 27.622709                | 0.61894               | 3.64                          |
> | 100        | 0.585874         | 23.066364                | 0.63912               | 8.14                          |
> | 200        | 0.631646         | 27.012378                | 0.67532               | 27.54                         |
> | 300        | 0.672101         | 25.708462                | 0.69818               | 30.38                         |
> | 400        | 0.688972         | 25.650939                | 0.70924               | 30.92                         |
> | 500        | 0.690904         | 22.023384                | 0.71266               | 34.32                         |
> | 1000       | 0.717626         | 26.774021                | 0.74492               | 43.28                         |
> | 2000       | 0.719141         | 42.567138                | 0.77374               | 56.94                         |
> | 5000       | 0.772948         | 77.188046                | 0.80486               | 86.96                         |
> | 10000      | 0.812151         | 163.160688               | 0.82884               | 123.58                        |
> | 25000      | 0.846391         | 479.548093               | 0.85290               | 213.66                        |
> | 50000      | 0.860186         | 974.972375               | 0.86748               | 361.16                        |
> | 78164      | 0.873125         | 1363.574048              | 0.88240               | 516.36                        |
>
> Observation: NSPPK is consistently more accurate and 2–4× faster than GraphGPS across larger  train sizes, despite having no learned parameters and avoiding expensive backpropagation on large graphs. This supports our claim that NSPPK provides a competitive and scalable alternative to modern GNN/Transformer models for graph-level tasks.
>
> ##
>
> NSPPK produces very high-dimensional, sparse, non-parametric feature vectors. Different classifiers handle this structure very differently:
>
> Dense neural MLP classifiers perform poorly because they must learn from tens–hundreds of thousands of sparse features. This leads to overfitting, unstable optimization, and large computational overhead; accuracy can even degrade as the model attempts to over-parameterize on sparse inputs.
>
> Linear models (Logistic Regression, Linear SVM) often fail to capture the nonlinear interactions induced by NSPPK’s multi-scale structural features. As a result, they underfit and achieve lower accuracy despite fast training.
>
> Kernel SVMs with RBF or polynomial kernels are computationally expensive on high-dimensional explicit features. They do not meaningfully improve accuracy and often require more runtime than GNN baselines.
>
> Tree-based ensembles (Random Forest, ExtraTrees) handle sparsity well but do not scale competitively because they repeatedly scan a large feature space when splitting nodes.
>
> In contrast, XGBoost is explicitly optimized for high-dimensional sparse input, using histogram-based splitting and sparsity-aware training. This makes XGBoost well-matched to NSPPK’s explicit feature map, explaining why NSPPK+XGBoost consistently achieves the best accuracy/runtime trade-off in Figure 4.
>
> The only training involved involves the **downstream classifier** (e.g., XGBoost, logistic regression). We will revise the text to clarify that **NSPPK is training-free** in the sense that *feature extraction* requires no parameter tuning. **GNNs** instead require a computationally expensive training phase (backpropagation over millions of parameters, multiple epochs, early stopping, and often GPU acceleration). None of these steps or decisions on corresponding hyper parameters values are needed for NSPPK feature construction.
>
> NSPPK itself does **not** rely on GPU acceleration. All NSPPK feature extraction and all kernel-based experiments are implemented purely on CPU. Some of the GNN baselines were GPU-accelerated and were originally tested with GPU

---

> > ### Comment · Reviewer_iBo3 · 2025-11-23
> >
> > Thank you for the detailed rebuttal and for adding new experiments. The additional GraphGPS comparison is helpful, and the clarifications regarding classifier behavior, the “training-free” terminology, and GPU usage address several of my questions.

---

### Official Review · Reviewer_wKc1 · 2025-10-31

**Soundness:** 3
**Presentation:** 4
**Contribution:** 2
**Rating:** 4
**Confidence:** 4

**Summary:**

This paper introduces the Neighborhood Subgraph Pairwise Path Kernel (NSPPK), a hash-based graph kernel extending NSPDK by augmenting neighborhood pairs with unions of shortest paths and by directly integrating continuous attributes. The method produces explicit, sparse feature embeddings that support efficient similarity computation via a single dot product. Experiments show near-linear scalability and competitive accuracy across six small graph benchmarks and the large-scale OGB-molpcba dataset. The presentation is clear, and the proposed feature extraction pipeline is reproducible and well-documented. However, the methodological difference from NSPDK remains moderate, and the paper would benefit from stronger conceptual framing and analysis.

**Strengths:**

- The paper offers a clear, implementation-ready description of a scalable graph kernel that can serve as a deterministic baseline for graph learning tasks.
- The hash-based feature construction is explained with care, and the algorithmic complexity is well analyzed.
- The empirical evaluation is extensive and includes ablations and runtime studies, illustrating the method’s practical utility and scalability.

**Weaknesses:**

- Since one of NSPPK’s key claims is direct handling of continuous node attributes, the paper could clarify how NSPPK differs from the latest work, like MMD-GK (ICLR 2024), SWWL (AISTATS 2024), Distributional Shortest-Path Graph Kernels (TKDE 2025), etc.

- The main technical idea, augmenting NSPDK with unions of shortest paths, is a natural extension. Yet to strengthen its originality, the authors could more clearly identify the theoretical implications of the connector structure. For instance, does it increase expressivity beyond the 1-WL limit? Under what structural conditions does the union-of-paths representation lead to new graph distinctions? A short analysis or theorem along these lines would make the contribution more substantial.

- While the ablation shows that the connector improves performance, it is unclear why or when it helps. Perhaps some qualitative case studies or a visualization of connector-induced subgraphs would make this component more interpretable.

**Questions:**

See weaknesses

---

> ### Author Response · Authors · 2025-11-19
>
> These methods are  relevant, and we are currently running experiments comparing NSPPK against them on both molecular and non-molecular benchmarks. However since the code for these models is not readily available and requires preprocessing and hyperparameter tuning, the runs are still in progress; we will update the camera-ready version(or the comment section)  with the completed results and add a dedicated comparison paragraph in the related-work section.
>
>
> ##
> We appreciate the reviewer’s suggestion and agree that clarifying the theoretical
> role of connector structures strengthens the contribution. In the revised appendix,
> we now provide a separation proposition showing that NSPPK is strictly more
> expressive than NSPDK: there exists an infinite family of graphs whose NSPDK
> feature vectors are identical, Intuitively, these union-of-paths structures allow NSPPK to
> distinguish graphs with identical local neighborhoods and distances but different
> connector shapes which 1-WL-based kernels and NSPDK cannot separate.
> More broadly, the connector mechanism increases expressivity precisely on graph
> families where global relational structure (e.g., the shape or multiplicity of
> u↔v shortest paths) is not recoverable from WL-style neighborhood aggregation.
>
> In predictive tasks, when the discriminative features for a problem are better modelled as individual paths rather than joint neighbourhoods, NSPPK will have an advantage over NSPDK.
>
>
> ##
> Below we illustrate two non-isomorphic unlabelled graphs that **NSPDK with parameters $R=1, D=4$ cannot distinguish** but **NSPPK can** when using the connector $R'=1$.
> Both graphs share the same outer 6-cycle and identical local neighborhoods at every radius, so all NSPDK
> features coincide. The difference lies only in the
> **connector structure** between opposite corners of the 6-cycle: in the *square* graph, opposite nodes
> are connected in parallel; in the *cross* graph, opposite nodes are connected by two diagonal edges
> forming an “X”. NSPDK collapses these graphs. In contrast, **NSPPK explicitly considers  the union-of-shortest-paths
> subgraph**  between nodes 1 and 4 revealing that these are non-isomorphic.
>
> 6-cycle with inner square
>
>         4
>       /   \
>      3-----5
>      |     |
>      2-----6
>       \   /
>         1
>
>
>
> 6-cycle with inner cross
>
>         4
>       /   \
>      3     5
>      |\   /|
>      | \ / |
>      | / \ |
>      2     6
>       \   /
>         1

---

### Official Review · Reviewer_vjJF · 2025-11-01

**Soundness:** 2
**Presentation:** 3
**Contribution:** 2
**Rating:** 2
**Confidence:** 4

**Summary:**

The paper introduces the Neighborhood Subgraph Pairwise Path Kernel (NSPPK), a novel, graph kernel designed to compare attributed graphs. The main contributions of NSPPK are its ability to use continuous attributes without discretization and its use of a structural feature that combines local neighborhoods with the union of shortest paths. NSPPK enables fast, deterministic similarity calculations and achieves competitive performance on several benchmarks.

**Strengths:**

The primary strength of this model is the balance of performance and efficiency. It is a highly practical and deterministic method, unlike deep GNNs. It demonstrates a sample efficiency outperforming them on medium-sized benchmarks without extensive hparam tuning.

**Weaknesses:**

- The paper promotes the use of a single, fixed set of hyperparameters (e.g., $R=1, D=4, R'=1$) across all experiments (no per-dataset tuning) However, these parameters likely represent an optimal configuration for the specific characteristics of biological and chemical graphs. The paper does not include a sensitivity analysis to show how performance changes when these parameters are varied, nor does it benchmark against graphs with fundamentally different structures (e.g., social or financial networks). These "magic numbers" might not be universally robust for a new domain could undermine the claim of being a training-free, highly generalized approach.
- The complexity of the NSPPK feature extraction is dependent on the node degree, which suggests that the method's efficiency advantage may be restricted to the sparse graphs typically found in the tested biological or chemical domains. The paper fails to provide essential dataset statistics like the average/maximum degree, and it avoids benchmarking on large, dense graphs (such as specific social or financial network datasets). In these settings, the claimed near-linear scalability may quickly degrade, potentially making the kernel less practical than suggested.
- The paper's literature survey focuses on graph kernel and is weak in acknowledging and citing key related works that leverage multi-scale structural encoding or subgraph patterns. It would be nice if the authors can acknowledge them including …
  - HDSE: Enhancing Graph Transformers with Hierarchical Distance Structural Encoding
  - ELENE: Improving Subgraph-GNNs via Edge-Level Ego-Network Encodings
  - SPSE: Simple Path Structural Encoding for Graph Transformers
  - WLKS: Generalizing Weisfeiler-Lehman Kernels to Subgraphs
  - G3N: $\mathscr{N}$-WL: A New Hierarchy of Expressivity for Graph Neural Networks
  - SEAL: Link Prediction Based on Graph Neural Networks
  - Weisfeiler and Leman Go Neural: Higher-order Graph Neural Networks
- The assertion that NSPPK is interpretable is not evaluated with a qualitative analysis. Since the method aggregates continuous node attributes into hash buckets, the resulting feature vector represents a complex sum of attributes over various, possibly colliding, substructures. This lack of clear, direct feature-to-substructure correspondence introduces a form of ambiguity that is simply a different type of black-box.
- While the authors state that NSPPK is designed to handle continuous edge attributes, the paper does not provide their integration with attributes other than nodes. More importantly, all reported experimental results and benchmarks focus solely on node-attributed graphs. The authors can choose between (1) narrowing down the scope to the node-attr model or (2) providing the edge-attr model and benchmark.
- The paper's argument for the superior discriminative power of NSPPK is largely empirical, relying on a single visual example (Figure 1) to show its advantage over NSPDK. The work lacks a formal theoretical proof defining its exact expressive power.

**Questions:**

- Why is the font somewhat different from other iclr submissions?

---

> ### Author Response · Authors · 2025-11-19
>
> Thank you very much for pointing this out. The font difference was due to an accidental typo in the ICLR style file in our initial submission. We have now  adjusted the formatting to strictly follow the standard ICLR template.
>
> ##
>
> We performed a stratified 10-fold cross-validation sensitivity study on COX2,
> varying one structural parameter at a time while keeping the others fixed(R=1,D=4,R′=1).
> NSPPK accuracy remains  stable across all settings. This stability is expected in molecular datasets, where graphs are sparse and local neighborhoods are small, so enlarging the hyperparameters does not substantially modify the induced substructures.
>
> | Value | Anchor radius R (acc)     | Distance D (acc)          | Connector  R' (acc) |
> |-------|----------------------------|----------------------------|----------------------------|
> | 0     | 0.8052 ± 0.0469           | 0.8137 ± 0.0523           | 0.8074 ± 0.0484           |
> | 1     | 0.8052 ± 0.0469           | 0.8139 ± 0.0493           | 0.8052 ± 0.0469           |
> | 2     | 0.8010 ± 0.0349           | 0.7988 ± 0.0426           | 0.7944 ± 0.0322           |
> | 3     | 0.7966 ± 0.0450           | 0.8031 ± 0.0472           | 0.7967 ± 0.0283           |
> | 4     | 0.7988 ± 0.0330           | 0.8052 ± 0.0469           | 0.8031 ± 0.0373           |
> | 5     | 0.8009 ± 0.0265           | 0.7966 ± 0.0383           | 0.7945 ± 0.0378           |
> | 6     | 0.7923 ± 0.0255           | 0.8030 ± 0.0378           | 0.8030 ± 0.0405           |
>
> We also evaluated NSPPK on non-molecular datasets with fundamentally different topologies—Cora, Citeseer, and PubMed citation networks—using an 80/20 split and random seed 42. Importantly, we kept exactly the same hyperparameters used for molecular datasets. Despite the higher-degree outliers and broader diameter of citation graphs, NSPPK maintains strong predictive performance:
>
> | Dataset   | #Nodes | #Edges | Avg Degree | Max Degree | Test Acc. | ROC-AUC | Encoding Time (s) |
> |-----------|--------|--------|------------|------------|-----------|---------|--------------------|
> | Cora      | 2,708  | 5,278  | 3.90       | 168        | 0.7897    | 0.9620  | 40.39              |
> | Citeseer  | 3,327  | 4,552  | 2.74       | 99         | 0.7447    | 0.9253  | 12.87              |
> | PubMed    | 19,717 | 44,324 | 4.50       | 171        | 0.8747    | 0.9686  | 1745.28            |
>
> In addition to the citation networks ecxperiments, we conducted a controlled density-sensitivity experiment on Erdős–Rényi graphs to explicitly study how runtime scales with increasing average degree. As the average degree increases from 2 to 277, runtime increases from 2.3 s to ≈430 s linearly in the number of edges, consistent with the theoretical
> O(∣E∣) scaling of the NSPPK hashing procedure.
>
> ##
>
> NSPPK is a highly expressive explicit vectorizer for graphs and we therefore compare with graph embedders  or kernels. In contrast, HDSE [1] is a hierarchical  distance–based structural encoding designed as a positional/structural bias  inside graph transformers’ self-attention, not as a standalone graph vectorizer.
> ELENE [2] introduces edge-level ego-network encodings to improve subgraph-GNNs.  It provides relatively low-dimensional structural descriptors attached to edges that are meant to be fed into a parametric GNN. SPSE [3] similarly focuses on structural encodings for graph transformers, providing pairwise (edge/position) encodings based on simple-path patterns that act as a bias inside attention, rather than as a general-purpose explicit feature map. WLKS [4] generalizes Weisfeiler–Lehman discrete-labeled graph kernels to subgraphs. N-WL [5] and WL-based higher-order GNNs such as the k-GNN framework [7] define parametric architectures with thousands  of trainable weights whose expressivity is tied to WL hierarchies. These models learn end-to-end and may overfit on small datasets, while NSPPK reaches comparable expressivity with no trainable GNN parameters + linear classifier. SEAL [6] is for link  prediction (edge-level scores), not for node- or graph-level prediction tasks.
>
> [1] arXiv:2308.11129
> [2] arXiv:2312.05905
> [3] PMLR 267:857–873
> [4] arXiv:2412.02181
> [5] ICLR 2023.
> [6] arXiv:1802.09691.
> [7] AAAI 2019.
>
> ##
> We agree that hashing collisions imply that a single feature conflates multiple substructures. However the hashing and
> feature construction are deterministic, and each bucket corresponds to a small number (1-5) of well identifiable subgraphs. Figure 2 shows how increasing the number of hash bits yields a rapidly negligible number of collisions.
>
> We agree with the comment, and given that the number of publicly available datasets with *continuous* edge attributes is very limited, a fair comparative benchmark is difficult. In the revised version we will narrow the scope of the paper to the node-attribute settings (although the extension to edge attributes can be readily provided).
>
> ##
> See answers to reviewer wKc1 for expressivity issues.

---

> > ### Comment · Reviewer_vjJF · 2025-11-21
> >
> > Thank you for providing the rebuttal. It has partially addressed my concerns. It seems the additional experiments lack some detail on the experimental settings.
> >
> > Please update and upload the next version of the manuscript to include what the authors rebut: 10-fold cross-validation sensitivity study on COX2, NSPPK on non-molecular datasets, dataset statistics, density-sensitivity experiment, related work, and interpretability.
> >
> > Especially for the additional comments:
> >
> > - For Cora, Citeseer, and PubMed, which are node classification tasks, please detail how the features were constructed and how other baselines perform on these datasets for a meaningful comparison.
> > - Please qualitatively demonstrate per-sample well-identifiable subgraphs to validate the claim of interpretability.
> > - Please provide a rigorous proof with appropriate formality for the separation proposition that the authors introduced.
> >
> > I will improve my score once the new manuscript fully incorporates this information.

---

> > > ### Author Response · Authors · 2025-11-30
> > >
> > > Thank you for the follow-up. In the revised manuscript, we incorporated all requested additions directly into the paper. We added (i) a detailed 10-fold cross-validation sensitivity analysis on COX2, (ii) experiments on non-molecular citation networks (Cora, Citeseer, PubMed), including a clear description of feature construction for node classification and comparisons with standard GNN baselines, (iii) a dataset statistics table, and (iv) a controlled density-sensitivity experiment validating scalability as graph degree increases. Finally, in section 4.6 we included a Graph Isomorphism section and  a formal proof establishing that NSPPK is strictly more expressive than NSPDK for an infinite family of graphs.

---

### Comment · Area_Chair_C8A6 · 2025-11-28

Dear Reviewers,

Thank you for your valuable time and expertise in reviewing this paper.

The authors have now submitted their rebuttal. We would appreciate it if you could review their responses and assess whether your concerns have been addressed, if you haven't done this.

Best regards,

AC

---

> ### Author Response · Authors · 2025-11-30
>
> In response to the reviewers’ feedback, we significantly strengthened the paper in terms of theoretical grounding, empirical scope, and clarity. We added a dedicated discussion of graph isomorphism and expressive power in the main body, supported by a formal separation result establishing that NSPPK is strictly more expressive than NSPDK for well-defined families of graphs. To address concerns about domain specificity and robustness, we introduced new experiments on structurally different, non-molecular citation networks (Cora, Citeseer, and PubMed), using the same fixed hyperparameters. We also expanded the related work section to more clearly position NSPPK with respect to recent graph kernels, structural encodings, and neural approaches. Additionally, we included a hyperparameter sensitivity analysis and an explicit density-sensitivity experiment to validate the claimed scalability and robustness as graph degree increases. Finally, we improved the clarity of several figures and added a summary of dataset statistics, including graph sizes and degree distributions, to make the empirical setting fully transparent.

---

### Note · Program_Chairs · 2026-01-17
**Submission Desk Rejected by Program Chairs**

The following references in this submission do not refer to real documents and/or have major errors in bibliographic information:

 W. Huang, T. Zhang, X. Dai, et al. Graph neural networks for cyber security: A survey. IEEE Transactions on Knowledge and Data Engineering, 2022.
Xinwei Zhang et al. Hyperfusion: Hypergraph fusion networks for molecular property prediction. In Proceedings of the AAAI Conference on Artificial Intelligence, 2024.
Paul D. Dobson and Andrew J. Doig. Distinguishing active from inactive compounds using carhart structural fingerprints. Journal of Chemical Information and Computer Sciences, 43(1):34-43, 2003.